# DuSA: Fast and Accurate Dual-Stage Sparse Attention Mechanism Accelerating Both Training and Inference

**Chong Wu**[*]
Dept of Electrical Eng
City University of Hong Kong
chongwu2-c@my.cityu.edu.hk
imroxaswc@gmail.com

**Jiawang Cao**[*]
Opus AI Research
gavin.cao@opus.pro

**Renjie Xu**[*]
Dept of Electrical Eng
City University of Hong Kong
harryxu950510@gmail.com

**Zhuoheng Ran**
Dept of Electrical Eng
City University of Hong Kong
zhuoheran2-c@my.cityu.edu.hk

**Maolin Che**[†]
School of Mathematics and Statistics
Guizhou University
chncml@outlook.com

**Wenbo Zhu**
Opus AI Research
vito.zhu@opus.pro

**Hong Yan**
Dept of Electrical Eng
City University of Hong Kong
h.yan@cityu.edu.hk

## Abstract

This paper proposes the Dual-Stage Sparse Attention (DuSA) mechanism for attention acceleration of transformers. In the first stage, DuSA performs intrablock sparse attention to aggregate local inductive biases. In the second stage, DuSA performs interblock sparse attention to obtain long-range dependencies. Both stages have low computational complexity and can be further accelerated by memory acceleration attention mechanisms directly, which makes DuSA faster than some extremely fast attention mechanisms. The dual-stage sparse attention design provides a lower error in approximating vanilla scaled-dot product attention than the basic single-stage sparse attention mechanisms and further advances the basic sparse attention mechanisms to match or even outperform vanilla scaled-dot product attention. Even in some plug and play situations, DuSA can still maintain low performance loss. DuSA can be used in both training and inference acceleration. DuSA achieves leading performance in different benchmarks: long range arena, image classification, semantic segmentation, object detection, text to video generation, and long context understanding, and accelerates models of different sizes.

## 1 Introduction

The vanilla scaled-dot product attention (VSA) mechanism [1] is a core technique in transformers due to its ability to capture and model long-range dependencies [2]. However, the softmax operation after the scaled-dot product becomes the main bottleneck that makes the computational complexity of VSA the square of the sequence length and significantly limits efficiency of VSA [3].

---

[*]Equal contribution

[†]Corresponding author

39th Conference on Neural Information Processing Systems (NeurIPS 2025).

To improve efficiency, various attention acceleration methods have been proposed. Among them, FlashAttention [4] is a milestone in attention acceleration that provides an exact method of attention acceleration by optimizing memory input/output (I/O) operations. Its successors [5, 6] further improve memory efficiency. In addition to these memory acceleration attention mechanisms, other methods focus on optimizing the computation bound by using the kernel trick to find a linear approximation [2, 3, 7–19] or utilizing the sparsity of attention matrices to perform local attention operations [18, 20–30]. These methods are usually called computation acceleration attention mechanisms. Memory acceleration attention mechanisms can usually provide an exact attention operation to replace the vanilla scaled-dot product attention mechanism directly. However, they are usually hardware-aware and their computational complexity is still quadratic with respect to the sequence length [18]. Computation acceleration attention mechanisms are usually faster than memory acceleration attention mechanisms, and some of them [18, 29, 30] can be further accelerated by memory acceleration attention mechanisms. However, most of them have inferior performance compared to VSA and cannot directly replace VSA without training or tuning.

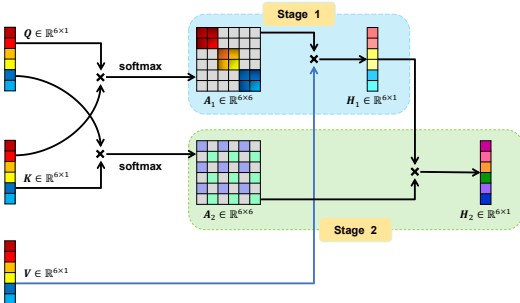

Figure 1: The overview of DuSA. Boxes with grey color denote zeros. Figs. 2 and 3 shows two examples for obtaining $A_2$.

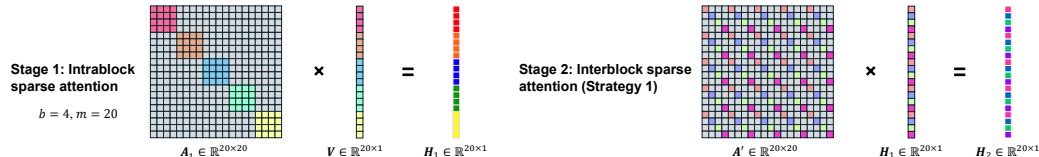

Figure 2: DuSA performs intrablock sparse attention within each block to obtain local inductive biases in Stage 1. DuSA performs interblock sparse attention to obtain global information in Stage 2. Here the blockify strategy is Strategy 1 in Remark 1. The different colors in attention matrices ($A_1$ and $A_2 = A'$) denote the different blocks. The color changes between $V$ and $H_1$ and between $H_1$ and $H_2$ denote the information aggregation process.

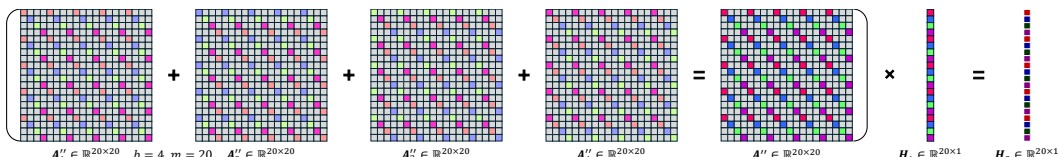

Figure 3: DuSA using Strategy 2 in interblock sparse attention can aggregate more information. $A_2 = A''$. $A_1'' = A'$ and $A_k''$ ($k \neq 1$) can be obtained using the rolling indices introduced in Remark 2.

This paper proposes Dual-Stage Sparse Attention (DuSA), a novel attention mechanism, for attention acceleration of transformers. This work builds on the basic single-stage sparse attention mechanisms [25] and advances the field further through the novel accurate dual-stage sparse attention mechanism design as shown in Fig. 1. DuSA performs intrablock sparse attention in the first stage and interblock sparse attention in the second stage, which makes DuSA capable of obtaining intrablock information (local inductive biases) and interblock information (long-range dependencies) at the same time and is significantly different from commonly used sparse attention mechanisms [23–25, 27–30]. Both

stages have low computational complexity and can be further accelerated by memory acceleration attention mechanisms directly. In summary, this paper has the following contributions.

(i) A novel and effective dual-stage sparse attention (DuSA) mechanism is proposed to significantly accelerate training and inference processes.

(ii) A mathematical relationship between DuSA and VSA is given: 1. The theoretical upper bound using DuSA to approximate VSA is derived. 2. The numerical results demonstrate that the dual-stage design has a lower approximation error than the single-stage design, despite the higher theoretical upper bound.

(iii) DuSA can approximate VSA with a low error: DuSA can maintain 97%+ performance using 75%-94% sparsity ratios by replacing VSA in vision transformers (ViTs) directly without training, and surpasses VSA after training.

(iv) DuSA surpasses state-of-the-art methods in terms of better performance and faster speed in different tasks.

## 2 Related work

Efficient attention mechanisms have been extensively studied to mitigate the quadratic complexity of attention. These methods can be classified into two categories based on their optimization focus: (1) memory acceleration methods and (2) computation acceleration methods.

### 2.1 Memory acceleration methods

Memory acceleration methods are usually hardware-aware and focus on minimizing memory I/O access to reduce complexity for some certain types of GPU architectures. FlashAttention [4] is the milestone for memory acceleration in attention computation. FlashAttention-2 (FA2) [5] is a major update version of FlashAttention, which further improves efficiency. FlashAttention-3 is a version that supports new features or characteristics of the new Hopper GPU architectures. Inspired by FlashAttention, some variants of FlashAttention have been proposed. FlashSigmoid [31] implements a memory efficient sigmoid attention mechanism based on FA2. Native sparse attention (NSA) [29] and SpargeAttn [30] implement memory efficient selective sparse attention based on FlashAttention. Different to NSA, SpargeAttn is designed only for inference acceleration, which cannot be used to accelerate the training process.

### 2.2 Computation acceleration methods

Computation acceleration methods can be classified into two types: (1) the linear kernelized attention mechanism; (2) the sparse/local attention mechanism according to whether or not the sparse property of the attention matrices is utilized.

#### 2.2.1 Linear kernelized attention mechanisms

Linear kernelized attention mechanisms use the kernel trick to approximate the softmax normalization of the vanilla scaled-dot product of the query matrix and the key matrix into a linear product of two independent feature maps. Powerful feature map design becomes the key problem in this kind of efficient attention mechanism. Rectified linear unit (ReLU) [32] based feature maps are commonly used to approximate softmax normalization of the vanilla scaled-dot product of the query matrix and the key matrix, such as the ReLU-based feature map dividing by sequence length [9], the cubic ReLU-based feature map [7], the convolution-enhanced ReLU-based feature map [33], and the exponential linear unit (ELU) [34] based feature map [10]. In addition to ReLU-based feature maps, there are some other types of feature maps, such as the *cos*-based nonlinear feature map [14] and the independent softmax-based feature map [8, 15, 18]. The attention matrix is approximately low-rank [17], therefore, another kind of feature map design is based on low-rank decomposition methods, such as the Nyström approximation [2, 3, 12, 35], the CUR decomposition [17], the singular value decomposition (SVD) [36], and matrix decomposition [37].

### 2.2.2 Sparse/local attention mechanisms

Sparse/local attention mechanisms are based on the sparse property of attention matrices and usually use blockify methods, such as sliding windows, to partition the original sequences into smaller blocks [18]. Attention computation is limited within these blocks. Computational complexity can be reduced to almost sublinear complexity with an increasing number of blocks. Blockify methods based on sliding windows were first introduced by Longformer [24] and Swin [20]. Due to the local nature of the blocks, global information is easily lost. Different compensation strategies have been proposed, and most existing methods are based on the combination of different sparse attention mechanisms. CSWin [21] introduces two different cross-shaped sliding windows in two parallel attention heads and combines the attention results of two heads to reduce global information loss. Some methods use several fixed-step or randomly selected tokens to introduce long-range dependencies [25, 27]. Reformer [28] proposes a locality-sensitive-hashing (LSH) algorithm for selecting tokens to perform sparse attention, while the explicit sparse transformer [23] proposes top-$k$ selective attention. NSA [29] combines compressed attention, top-$k$ selective attention, and sliding window-based attention to propose a hierarchical sparse attention mechanism to obtain different levels of information. Unlike these hierarchical sparse attention mechanisms, ELFATT [18] proposes a hybrid head architecture which combines simple sparse blockify attention with global kernelized linear attention in two parallel heads to obtain global information. XAttention [38] uses the sum of antidiagonal values of the attention matrix to select blocks for a further sparse attention computation within these blocks.

This work is significantly different from the above sparse attention mechanisms. DuSA introduces a novel dual-stage sparse attention mechanism design as shown in Fig. 1. The first stage (intrablock sparse attention) is used to obtain intrablock information (local inductive biases) and the second stage (interblock sparse attention) is used to aggregate all blocks according to their similarities (attention scores) to obtain interblock information (long-range dependencies). Hence, DuSA can approximate VSA with a low error. Both stages have low computational complexity and are compatible with memory efficient methods, which can be further accelerated. As an enhanced alternative for basic sparse attention, DuSA can further improve and accelerate some advanced sparse attention mechanisms.

## 3 Methods

### 3.1 Vanilla scaled-dot product attention

For an input embedding matrix $\boldsymbol{H} \in \mathbb{R}^{m \times c}$, after three linear transformations, we can obtain the following,

$$\boldsymbol{Q} = \boldsymbol{H}\mathcal{W}_Q, \quad \boldsymbol{K} = \boldsymbol{H}\mathcal{W}_K, \quad \boldsymbol{V} = \boldsymbol{H}\mathcal{W}_V, \tag{1}$$

where $\mathcal{W}_Q \in \mathbb{R}^{c \times c}$, $\mathcal{W}_K \in \mathbb{R}^{c \times c}$, and $\mathcal{W}_V \in \mathbb{R}^{c \times c}$ are three scaling transformation matrices, $\boldsymbol{Q} \in \mathbb{R}^{m \times c}$, $\boldsymbol{K} \in \mathbb{R}^{m \times c}$, and $\boldsymbol{V} \in \mathbb{R}^{m \times c}$ are the query matrix, key matrix, and value matrix, respectively. Vanilla scaled-dot product attention (VSA) obtains the attention matrix $\boldsymbol{A} \in \mathbb{R}^{m \times m}$ and the updated embedding matrix $\boldsymbol{H}$ as follows,

$$\boldsymbol{A} = \mathrm{softmax}\left(\frac{\boldsymbol{Q}\boldsymbol{K}^{\top}}{\sqrt{c}}\right), \quad \boldsymbol{H} \leftarrow \boldsymbol{A}\boldsymbol{V}. \tag{2}$$

Since $m \gg c$, the softmax operation for the scaled-dot product of $\boldsymbol{Q}$ and $\boldsymbol{K}$ makes VSA quadratic computational complexity with respect to $m$ and becomes the main bottleneck of efficiency.

### 3.2 Dual-stage sparse attention

Dual-Stage Sparse Attention (DuSA) can be expressed as a two-stage sparse attention process as Figs. 2 and 3 show. In the first stage, DuSA performs local attention (intrablock sparse attention) within each block as follows,

$$\boldsymbol{A}_1 = \mathrm{softmax}\left(\frac{\boldsymbol{Q}\boldsymbol{K}^{\top}}{\sqrt{c}} \odot \boldsymbol{Z}_1\right), \quad \boldsymbol{H}_1 \leftarrow \boldsymbol{A}_1\boldsymbol{V}, \tag{3}$$

where $\odot$ denotes the Hadamard product [39] and $\boldsymbol{Z}_1 = \boldsymbol{D}_{(m/b)} \otimes \boldsymbol{U}_{(b)}$ with $\otimes$ denoting the Kronecker product [39], $b$ being the block size, $\boldsymbol{D}_{(m/b)} \in \mathbb{R}^{(m/b) \times (m/b)}$ being a matrix of which diagonal elements are 1 and all the other elements are $-\infty$, and $\boldsymbol{U}_{(b)} \in \mathbb{R}^{b \times b}$ being the all-ones matrix.

Before considering the second stage of DuSA, we need to consider the mathematical expressions for blockify Strategies 1 and 2 in Figs. 2 and 3.

*Remark* 1. Let $\mathbb{J}_j = \{j, j+b, \ldots, j+(m/b-1)b\}$ with $j = 1, 2, \ldots, b$. Hence, for blockify Strategy 1, the matrix $\boldsymbol{Z}_2 \in \mathbb{R}^{m \times m}$ is defined as $\boldsymbol{Z}_2(\mathbb{J}_j, \mathbb{J}_j)^1 = \boldsymbol{U}_{(m/b)}$ with $j = 1, 2, \ldots, b$, and all the other elements of $\boldsymbol{Z}_2$ are $-\infty$. Therefore, one has

$$\boldsymbol{A}' = \text{softmax}\left(\frac{\boldsymbol{Q}\boldsymbol{K}^\top}{\sqrt{c}} \odot \boldsymbol{Z}_2\right), \quad \boldsymbol{H}' \leftarrow \boldsymbol{A}'\boldsymbol{H}_1. \tag{4}$$

*Remark* 2. We also introduce several notations (rolling indices) as follows:

$$\{\mathbb{J}_1^{(1)}, \mathbb{J}_2^{(1)}, \ldots, \mathbb{J}_{b-1}^{(1)}, \mathbb{J}_b^{(1)}\} = \{\mathbb{J}_1, \mathbb{J}_2, \ldots, \mathbb{J}_b\},$$
$$\{\mathbb{J}_1^{(2)}, \mathbb{J}_2^{(2)}, \ldots, \mathbb{J}_{b-1}^{(2)}, \mathbb{J}_b^{(2)}\} = \{\mathbb{J}_2, \ldots, \mathbb{J}_b, \mathbb{J}_1\},$$
$$\cdots$$
$$\{\mathbb{J}_1^{(b)}, \mathbb{J}_2^{(b)}, \ldots, \mathbb{J}_{b-1}^{(b)}, \mathbb{J}_b^{(b)}\} = \{\mathbb{J}_b, \mathbb{J}_1, \ldots, \mathbb{J}_{b-1}\}.$$

For each $k$, the matrix $\boldsymbol{A}_k''$ is defined as $\boldsymbol{A}_k''(\mathbb{J}_j, \mathbb{J}_j) = \boldsymbol{A}(\mathbb{J}_j^{(k)}, \mathbb{J}_j^{(k)})$ with $j = 1, 2, \ldots, b$, and $\boldsymbol{A}_k''(\mathbb{J}_j, \mathbb{J}_{j'}) = \boldsymbol{0}_{(m/b)}$ with $\boldsymbol{0}_{(m/b)} \in \mathbb{R}^{(m/b) \times (m/b)}$ being the all-zeros matrix and $j \neq j'$. Note that $\boldsymbol{A}_1'' = \boldsymbol{A}'$. Therefore, for blockify Strategy 2, we have

$$\boldsymbol{A}'' = \sum_{k=1}^{b} \boldsymbol{A}_k'', \quad \boldsymbol{H}'' \leftarrow \boldsymbol{A}''\boldsymbol{H}_1. \tag{5}$$

The upper bounds of Eqs. (3-5) are discussed in Section A1 in Appendix. Based on Remarks 1 and 2, the second stage of DuSA performs interblock sparse attention as follows,

$$\boldsymbol{A}_2 = \boldsymbol{A}'', \quad \boldsymbol{H}_2 \leftarrow \boldsymbol{H}'' = \boldsymbol{A}''\boldsymbol{H}_1 = \boldsymbol{A}_2(\boldsymbol{A}_1\boldsymbol{V}), \tag{6}$$

or

$$\boldsymbol{A}_2 = \boldsymbol{A}', \quad \boldsymbol{H}_2 \leftarrow \boldsymbol{H}' = \boldsymbol{A}'\boldsymbol{H}_1 = \boldsymbol{A}_2(\boldsymbol{A}_1\boldsymbol{V}). \tag{7}$$

## 4 Experiments and results

### 4.1 Experiment settings

DuSA is evaluated in image classification (ImageNet-1K [40]), long range arena (LRA [41]), text to video generation (Open-Sora 2.0 [42] prompt sets[2]), semantic segmentation (ADE20K [43]), object detection (MS COCO 2017 [44]), and long context understanding (LongBench [45]) tasks. DuSA0 denotes DuSA (without using Strategies 1 and 2), DuSA1 denotes DuSA (Strategy 1), and DuSA2 denotes DuSA (Strategy 2). If not specified, all results of DuSA were obtained using DuSA2. For image classification, semantic segmentation, and object detection tasks, we compared DuSA with Agent [16], cross-shaped sliding window attention (CSW) [21], EFFATT [15], ELFATT [18], FA2 [5], FLatten [7], sliding window attention (SW) [20], and VSA [1]. The ViT backbones used for image classification, semantic segmentation, and object detection tasks are: CSWin-T/B [7, 21] and Swin-T/B [20]. The original backbones using CSW or SW are named with the suffix "LOCAL": CSWin-T/B-LOCAL and Swin-T/B-LOCAL. The naming scheme for backbones using other attention mechanisms to replace CSW or SW is to add the corresponding attention mechanism name as a suffix after the backbone name, for example, CSWin-T-DuSA. We followed the training protocol of [18, 21, 46] to adopt the same training settings and data augmentation methods to train all methods. Some advanced non-ViT architectures: ConvNeXt-T [47] and VMamba-T [46] are selected for comparison. The experiments were conducted using 8 NVIDIA vGPU (32GB) GPUs. For LRA, we compared DuSA with FA2, Linformer [22], Nyströmformer [3], Primal [36], Reformer [28], and VSA. We followed the training protocol and the settings of [18, 36] and performed the experiments using 1

---

[1]For a given matrix $\boldsymbol{B} \in \mathbb{R}^{m \times n}$, and two index sets $\mathbb{I} = \{i_1, \ldots, i_s\} \subset \{1, 2, \ldots, m\}$ with $1 \leq i_1 < \cdots < i_s \leq m$ and $\mathbb{J} = \{j_1, \ldots, j_t\} \subset \{1, 2, \ldots, n\}$ with $1 \leq j_1 < \cdots < j_t \leq n$, we let $\boldsymbol{B}(\mathbb{I}, \mathbb{J})$ denote the submatrix $\in \mathbb{R}^{s \times t}$ of $\boldsymbol{B}$ with entries $\{b_{ij}\}_{i \in \mathbb{I}, j \in \mathbb{J}}$.

[2]https://github.com/hpcaitech/Open-Sora/blob/main/assets/texts

NVIDIA A100 (40GB) GPU. We followed the settings of [30] to run text to video generation using DuSA. The backbone used is CogVideoX-2B [48]. We compared DuSA with FA2 and SpargeAttn [30]. Complexity analysis, the ablation study about different block sizes, and more experiments (plug and play examples, the use of DuSA in vanilla ViT architectures [49], memory consumption patterns, and the comparison of DuSA with FlashAttention of FlashInfer [50] and XAttention [38] in accelerating the prefilling stage of Llama-3.1-8B-Instruct [51] on LongBench) are available in Appendix. The source code of DuSA is available in Supplementary Material. For ImageNet-1K, ADE20K and MS COCO 2017, all methods including DuSA that are compatible with FA2, are accelerated by FA2. For text to video generation, DuSA is further accelerated by SpargeAttn.

Table 1: The test accuracy comparison of different methods on LRA. The best values are in bold.

| Dataset (sequence length) | Linformer | Nyströmformer | Primal | Reformer | VSA | DuSA |
|---|---|---|---|---|---|---|
| ListOps (2K) | 37.3 | 37.2 | 37.3 | 19.1 | 37.1 | **38.1** |
| Text (4K) | 55.9 | **65.5** | 61.2 | 64.9 | 65.0 | 65.3 |
| Retrieval (4K) | 79.4 | 79.6 | 77.8 | 78.6 | 79.4 | **81.4** |
| Image (1K) | 37.8 | 41.6 | 43.0 | **43.3** | 38.2 | 42.7 |
| Pathfinder (1K) | 67.6 | 70.9 | 68.3 | 69.4 | **74.2** | 73.3 |
| Average accuracy (%) | 55.6 | 59.0 | 57.5 | 55.1 | 58.8 | **60.2** |

Table 2: The comparison of running time (s) per 1K training steps and peak memory consumption of different methods on LRA using 1 NVIDIA H20 (96GB) GPU.

| Dataset (sequence length) | FA2 | Linformer | Nyströmformer | Primal | Reformer | VSA | DuSA |
|---|---|---|---|---|---|---|---|
| | Time (s) \| Memory (GB) | | | | | | |
| ListOps (2K) | 24.2 \| **0.4** | 19.6 \| 1.1 | 28.3 \| 0.6 | 21.2 \| 0.5 | 34.9 \| 1.4 | 71.2 \| 4.3 | **16.8** \| 0.5 |
| Text (4K) | 63.7 \| **0.8** | 35.1 \| 2.2 | 50.4 \| 1.2 | 33.9 \| 0.9 | 69.3 \| 2.8 | 263.2 \| 16.5 | **30.7** \| 1.3 |
| Retrieval (4K) | 126.3 \| **1.4** | 69.1 \| 4.1 | 96.0 \| 2.4 | 66.5 \| 1.9 | 137.8 \| 5.4 | 523.0 \| 17.2 | **66.1** \| 2.2 |
| Image (1K) | 48.9 \| **0.8** | 45.8 \| 2.2 | 69.7 \| 1.4 | 49.8 \| 1.0 | 82.9 \| 2.9 | 109.2 \| 4.5 | **44.8** \| 1.2 |
| Pathfinder (1K) | 63.8 \| **0.8** | 63.4 \| 2.2 | 91.0 \| 1.4 | 64.6 \| 1.0 | 113.5 \| 2.9 | 144.2 \| 4.5 | **52.2** \| 1.1 |

## 4.2 Long range arena

DuSA is compared with state-of-the-art attention mechanisms in LRA as Table 1 shows. It can be seen that DuSA achieves state-of-the-art performance in LRA. DuSA significantly outperforms VSA in long hierarchically structured data modeling (ListOps (2K)), text classification (Text (4K)), image classification on sequences of pixels (Image (1K)), and document retrieval (Retrieval (4K)) tasks. Only in long-range spatial dependency learning (Pathfinder (1K)), DuSA is slightly inferior to VSA, but it is significantly better than other efficient attention mechanisms in this task. Furthermore, DuSA is faster than all comparison methods and offers 2.4-8.6× speedups over VSA and 1.1-2.1× speedups over FA2 and achieves the third lowest memory consumption as shown in Table 2.

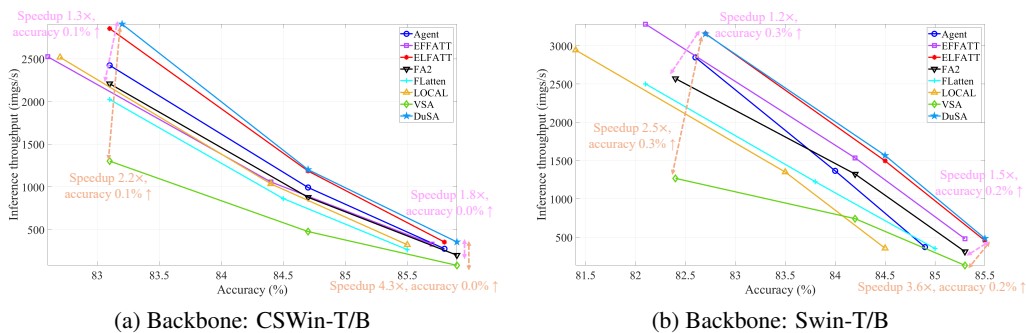

(a) Backbone: CSWin-T/B          (b) Backbone: Swin-T/B

Figure 4: Accuracy-efficiency (inference throughput) curves of different methods on ImageNet-1K.

## 4.3 Image classification performance

Fig. 4 presents a comprehensive performance comparison of various methods in the image classification task (ImageNet-1K). The experimental results demonstrate that DuSA achieves superior

classification accuracy compared to all methods in different model sizes and input resolutions. In addition, DuSA is faster than all comparison methods and offers 2.2-4.3× speedups over VSA and 1.2-1.8× speedups over FA2 using CSWin/Swin backbones. More detailed results can be found in Table A1 in Appendix. Fig. 5 shows the visual comparison of class activation maps (CAMs) of DuSA, ELFATT, and VSA. Compared to ELFATT and VSA, CAMs of DuSA (Strategy 2) are more accurate than theirs. DuSA2 focuses on the groudtruth objects more accurately than DuSA0/DuSA1.

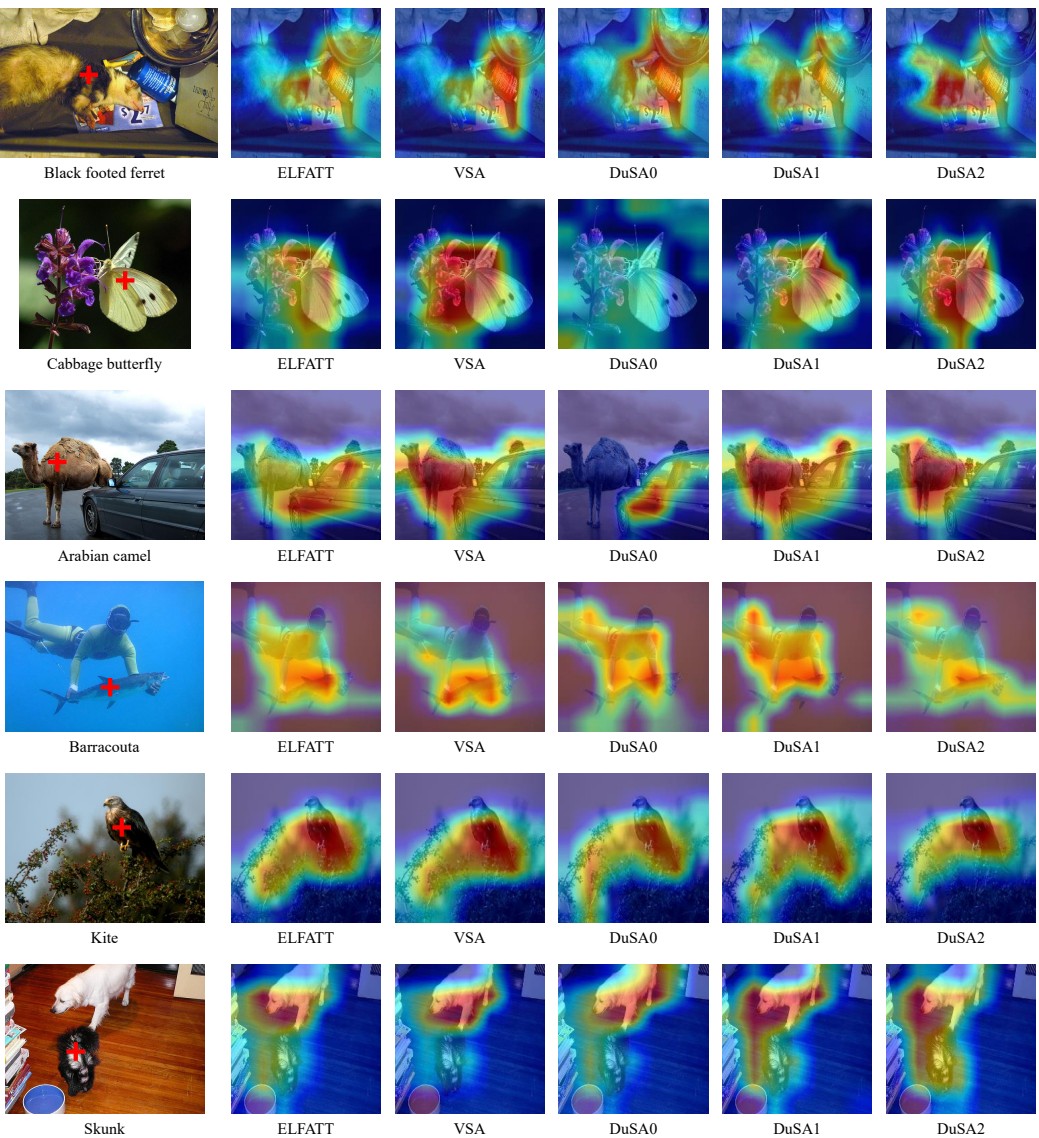

Figure 5: Class activation maps (CAMs) comparison of CSWin-T-DuSA, CSWin-T-ELFATT, and CSWin-T-VSA using Score-CAM [52]. Note: CAMs obtained by DuSA2 and DuSA1 focus on the groudtruth objects more accurately than those of DuSA0 which further demonstrates the effectiveness of the dual-stage spase attention design. In most cases, CAMs of DuSA2 cover more area of the groudtruth objects and less area of the background than those of DuSA1. Compared to ELFATT and VSA, CAMs of DuSA2 are even more accurate than theirs.

## 4.4 Semantic segmentation performance

Table 3 presents a comprehensive performance comparison of various methods on ADE20K. DuSA significantly outperforms all comparison methods in terms of mean class accuracy (mAcc) and mean intersection over union (mIoU). As for inference speed, DuSA matches the speed of some extremely

Table 3: ADE20K semantic segmentation comparison. Note: Number of floating point operations (FLOPs) are based on an input size of $512 \times 2048$. # is parameter numbers.

| Method | mAcc | mIoU | FPS (imgs/s) | # | FLOPs |
|---|---|---|---|---|---|
| UperNet [53] using 160K fine-tuning iterations. Inference throughput (FPS) was measured on 1 NVIDIA H20 with a batch size of 1 and mixed precision. | | | | | |
| CSWin-T | Swin-T | | | | | |
| Agent | 60.8 \| 58.5 | 48.5 \| 46.7 | 17 \| 4 | 50M \| 61M | 929.64G \| 957.50G |
| EFFATT | 60.6 \| 58.3 | 48.8 \| 46.7 | 33 \| 39 | 50M \| 60M | 930.34G \| 939.35G |
| ELFATT | 61.2 \| 59.3 | **49.6** \| 47.7 | 32 \| 38 | 50M \| 62M | 929.53G \| 943.94G |
| FLatten | **61.4** \| 57.0 | 49.3 \| 44.8 | 27 \| 35 | 51M \| 61M | 930.19G \| 944.62G |
| LOCAL | 61.1 \| 55.6 | **49.6** \| 44.5 | 28 \| 38 | 50M \| 60M | 928.68G \| 945.66G |
| VSA | 61.1 \| 59.3 | 48.8 \| 47.8 | 6/14 (FA2) \| 5/14 (FA2) | 50M \| 60M | 2458.75G/928.67G (FA2) \| 2873.79G/937.84G (FA2) |
| DuSA | **61.4** \| **59.8** | **49.6** \| **48.0** | 32 \| 37 | 50M \| 62M | 928.69G \| 942.98G |
| Others | | | | | |
| ConvNeXt-T | 58.3 | 46.1 | 37 | 60M | 939.69G |
| VMamba-T | 59.3 | 47.9 | 34 | 62M | 948.78G |

fast methods: EFFATT and ELFATT, and offers 5.3-7.4× speedups over VSA and 2.3-2.6× speedups over FA2. Both backbones using DuSA significantly outperform ConvNeXt-T and VMamba-T. Swin-T-DuSA is as fast as ConvNeXt-T and faster than VMamba-T in semantic segmentation.

## 4.5 Object detection performance

Table 4 compares the object detection performance of various methods on MS COCO 2017. DuSA matches the state-of-the-art performance of ELFATT and significantly outperforms other attention mechanisms with higher box/mask average precision ($AP^b/AP^m$) in both '1×' fine-tuning training schedule and '3×' multiscale (MS) fine-tuning training schedule. DuSA offers 5.6-8.3× speedups over VSA and 2.2-2.9× speedups over FA2. CSWin-T-DuSA significantly outperforms VMamba-T in object detection with higher box/mask average precision and faster speed. Swin-T-DuSA also significantly outperforms ConvNeXt-T in terms of both performance and speed.

Table 4: MS COCO 2017 object detection comparison. Note: FLOPs are based on an input size of $1280 \times 800$. FPS was measured on 1 NVIDIA H20 with a batch size of 1 and mixed precision.

| Method | $AP^b$ | $AP^b_{50}$ | $AP^b_{75}$ | $AP^m$ | $AP^m_{50}$ | $AP^m_{75}$ | FPS (imgs/s) | # | GFLOPs |
|---|---|---|---|---|---|---|---|---|---|
| Mask R-CNN [54] 1× schedule \| 3×MS schedule | | | | | | | | | |
| CSWin-T | | | | | | | | | |
| Agent | 46.8 \| 49.3 | 68.9 \| 70.8 | 51.3 \| 53.9 | 42.3 \| 43.9 | 65.9 \| 67.9 | 45.3 \| 47.3 | 20 | 40M | 254.51 |
| EFFATT | 46.1 \| 48.5 | 68.3 \| 70.0 | 50.5 \| 53.2 | 41.9 \| 43.4 | 65.5 \| 67.3 | 45.3 \| 46.9 | 36 | 40M | 255.20 |
| ELFATT | **47.0** \| 49.4 | **69.2** \| 70.9 | 51.4 \| **54.4** | 42.6 \| 44.0 | 66.4 \| **68.0** | **45.9** \| 47.5 | 33 | 40M | 254.40 |
| FLatten | 46.6 \| 48.9 | 68.8 \| 70.8 | 51.0 \| 53.5 | 42.2 \| 43.9 | 65.7 \| 67.9 | 45.3 \| 47.3 | 28 | 41M | 255.05 |
| LOCAL | 46.5 \| 49.3 | 68.5 \| 70.8 | 51.0 \| 54.3 | 42.1 \| 44.0 | 65.6 \| 67.8 | 45.3 \| 47.5 | 28 | 40M | 253.57 |
| VSA | **47.0** \| 48.8 | 69.1 \| 70.0 | **51.9** \| 53.5 | 42.6 \| 43.6 | 66.1 \| 67.4 | **45.9** \| 47.1 | 7/18 (FA2) | 40M | 1712.76/253.56 (FA2) |
| DuSA | 46.9 \| **49.4** | **69.2** \| **70.9** | 51.7 \| **54.4** | **42.7** \| **44.1** | **66.4** \| 67.8 | **45.9** \| **47.6** | 39 | 40M | 253.58 |
| Swin-T | | | | | | | | | |
| Agent | 44.6 \| 47.3 | 67.5 \| 69.5 | 48.7 \| 51.9 | 40.7 \| 42.7 | 64.4 \| 66.4 | 43.4 \| 46.2 | 5 | 48M | 278.42 |
| EFFATT | 44.7 \| 47.6 | 67.0 \| 69.4 | 48.9 \| 52.6 | 41.1 \| 42.7 | 64.0 \| 65.9 | 44.4 \| 46.1 | 46 | 48M | 261.95 |
| ELFATT | **46.1** \| **48.5** | **68.3** \| 70.4 | 50.8 \| **53.4** | 42.1 \| 43.6 | **65.4** \| 67.3 | 45.3 \| 47.3 | 45 | 50M | 266.43 |
| FLatten | 44.2 \| 46.5 | 67.3 \| 68.5 | 48.5 \| 50.8 | 40.2 \| 42.1 | 63.8 \| 65.4 | 43.0 \| 45.1 | 41 | 49M | 266.43 |
| LOCAL | 42.7 \| 46.0 | 65.2 \| 68.1 | 46.8 \| 50.3 | 39.3 \| 41.6 | 62.2 \| 65.1 | 42.2 \| 44.9 | 45 | 48M | 267.01 |
| VSA | 45.4 \| 48.0 | 67.9 \| 70.0 | 49.7 \| 52.7 | 41.6 \| 43.3 | 65.0 \| 67.0 | 44.8 \| 46.8 | 6/17 (FA2) | 48M | 2106.75/260.48 (FA2) |
| DuSA | 46.0 \| 48.4 | 68.2 \| 70.3 | **50.9** \| 53.0 | **42.2** \| 43.8 | 65.3 \| **67.5** | **45.5** \| **47.5** | 50 | 50M | 265.50 |
| Other tiny models | | | | | | | | | |
| ConvNeXt | 44.2 \| 46.2 | 66.6 \| 67.9 | 48.3 \| 50.8 | 40.1 \| 41.7 | 63.3 \| 65.0 | 42.8 \| 44.9 | 44 | 48M | 262.29 |
| VMamba | 47.4 \| 48.9 | 69.5 \| 70.6 | 52.0 \| 53.6 | 42.7 \| 43.7 | 66.3 \| 67.7 | 46.0 \| 46.8 | 35 | 50M | 271.16 |

Table 5: The comparison of text to video generation using CogVideoX-2B (sequence length: 17K) as the backbone. Note: The speed metric, TOPS (tera operations per second) [30], is the attention kernel speed. The speed was obtained using 1 NVIDIA vGPU (32GB) with BF16 precision.

| Method | TOPS ↑ | CLIPSIM ↑ | CLIP-T ↑ | VQA-a ↑ | VQA-t ↑ | Flow-score ↑ |
|---|---|---|---|---|---|---|
| FA2 | 103 | 0.2022 | **0.9971** | 51.6921 | 48.3386 | 5.6719 |
| SpargeAttn (sparsity: 0.46) | 198 | 0.2015 | 0.9968 | 51.5676 | 48.2785 | 5.7631 |
| DuSA (sparsity: 0.50) | **258** | **0.2024** | 0.9966 | 51.6049 | 48.3379 | **6.0482** |

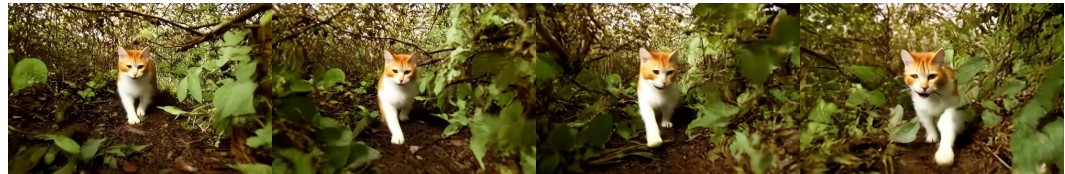

(a) FA2 (CLIPSIM: 0.2161, CLIP-T: 0.9983, VQA-a: 53.4576, VQA-t: 48.9721, Flow-score: 12.3805)

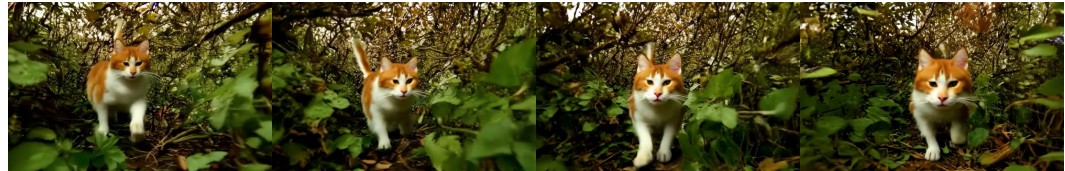

(b) SpargeAttn (CLIPSIM: 0.2109, CLIP-T: 0.9984, VQA-a: 52.9862, VQA-t: 48.3282, Flow-score: 10.4248)

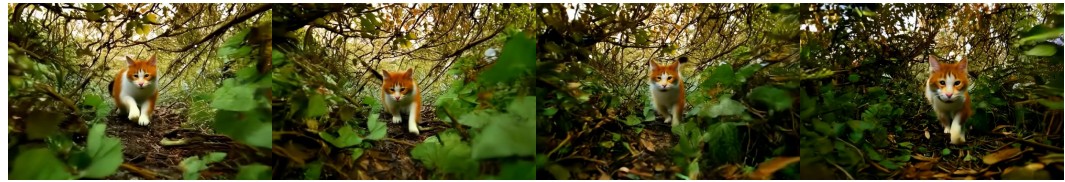

(c) DuSA (CLIPSIM: 0.2108, CLIP-T: 0.9986, VQA-a: 53.5017, VQA-t: 48.7888, Flow-score: 8.7635)

Figure 6: DuSA achieves noninferior video generation quality compared to VSA using FA2 for acceleration and outperforms SpargeAttn.

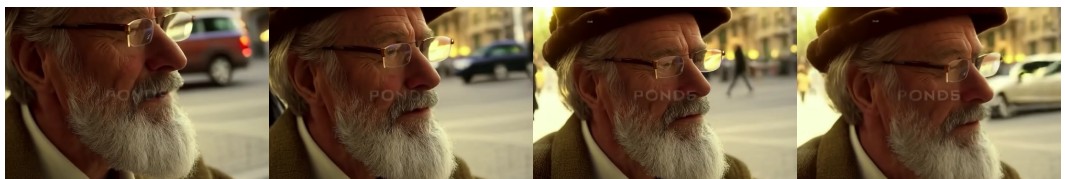

(a) FA2 (CLIPSIM: 0.1955, CLIP-T: 0.9992, VQA-a: 54.6491, VQA-t: 49.1951, Flow-score: 6.6526)

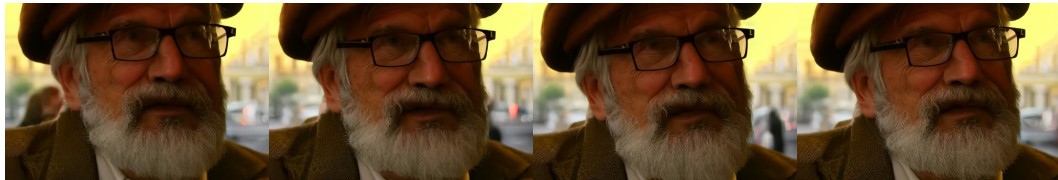

(b) SpargeAttn (CLIPSIM: 0.2044, CLIP-T: 0.9991, VQA-a: 53.2634, VQA-t: 48.8170, Flow-score: 6.1444)

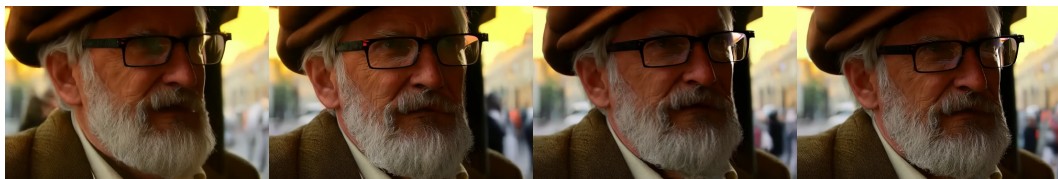

(c) DuSA (CLIPSIM: 0.2065, CLIP-T: 0.9982, VQA-a: 53.5948, VQA-t: 49.1732, Flow-score: 6.7921)

Figure 7: DuSA achieves noninferior video generation quality compared to VSA using FA2 for acceleration and outperforms SpargeAttn.

## 4.6 Text to video generation performance

Table 5 compares DuSA, FA2, and SpargeAttn in text to video generation (Open-Sora 2.0 prompt sets) using CogVideoX-2B as the backbone. We followed [30] to use CLIPSIM [55], CLIP-T [55], VQA-a [56], VQA-t [56], and Flow-score [57] to evaluate text-video alignment, aesthetic and technical quality of the video, and temporal consistency. Using a similar sparsity ratio, DuSA obtains better text-video alignment (CLIPSIM), higher video aesthetic and technical quality (VQA-a and VQA-t), and better temporal consistency (Flow-score) than SpargeAttn and its performance is almost at the same level compared to VSA using FA2 for acceleration. DuSA is faster than FA2 and SpargeAttn using BF16 precision. DuSA uses 98s, while FA2 uses 147s and SpargeAttn uses 106s on 1 NVIDIA vGPU (32GB) with BF16 precision for end-to-end generation. Figs. 6 and 7 show the visual comparison of DuSA, SpargeAttn, and VSA (FA2) in text to video generation. DuSA achieves better video generation quality than SpargeAttn and noninferior quality compared to VSA.

Table 6: Effect of different blockify strategies or stage execution orders of DuSA on ImageNet-1K. Note: FPS was measured on 1 NVIDIA H20 using a batch size of 512 and mixed precision.

| Method | Resolution | Accuracy (%) | FPS (imgs/s) | # | FLOPs |
|---|---|---|---|---|---|
| CSWin-T | | | | | |
| DuSA0 | $224^2$ | 82.9 | 3197 | 20M | 4.09G |
| DuSA1 (1st: Intrablock; 2nd: Interblock) | $224^2$ | 83.0 | 2909 | 20M | 4.09G |
| DuSA1 (1st: Interblock; 2nd: Intrablock) | $224^2$ | 82.9 | 2880 | 20M | 4.09G |
| DuSA2 (1st: Intrablock; 2nd: Interblock) | $224^2$ | 83.2 | 2908 | 20M | 4.09G |
| DuSA2 (1st: Interblock; 2nd: Intrablock) | $224^2$ | 83.0 | 2866 | 20M | 4.09G |
| Swin-T | | | | | |
| DuSA0 | $224^2$ | 82.5 | 3481 | 30M | 4.63G |
| DuSA1 (1st: Intrablock; 2nd: Interblock) | $224^2$ | 82.6 | 3136 | 30M | 4.63G |
| DuSA1 (1st: Interblock; 2nd: Intrablock) | $224^2$ | 82.6 | 3095 | 30M | 4.63G |
| DuSA2 (1st: Intrablock; 2nd: Interblock) | $224^2$ | 82.7 | 3154 | 30M | 4.63G |
| DuSA2 (1st: Interblock; 2nd: Intrablock) | $224^2$ | 82.6 | 3062 | 30M | 4.63G |

## 4.7 Ablation studies about different blockify strategies and different stage execution orders

Table 6 shows the comparison of DuSA using different blockify strategies on ImageNet-1K. Without using any blockify strategies, the speed is the fastest; however, its performance is also the lowest. Strategy 2 (DuSA2) obtains the highest accuracy because it includes more token information for each token in token mixing, as Fig. 3 shows. And its speed is almost the same as that of Strategy 1. Additionally, as illustrated in the visual comparison of Fig. 5, CAMs obtained using Strategies 1 and 2 focus more accurately on the ground-truth objects than those obtained without using Strategies 1 and 2, further demonstrating the effectiveness of the dual-stage sparse attention design. In most cases, CAMs of Strategy 2 cover more area of the groudtruth objects and less area of the background than those of Strategy 1. Table 6 also shows the comparison of DuSA using different execution orders of two stages on ImageNet-1K. Due to the change in execution order, the speed is slightly different due to the different reshaping operations and convolutions for different value matrices used for position encoding. Performing intrablock sparse attention first usually outperforms performing interblock sparse attention first in terms of both speed and accuracy.

## 5 Conclusions

In this paper, we propose DuSA for attention acceleration of transformers. DuSA further advances single-stage sparse attention mechanisms through the novel dual-stage sparse attention mechanism design. Both stages are designed to have low computational complexity and can be further accelerated by memory acceleration methods. The dual-stage sparse attention design provides a lower error in approximating vanilla scaled-dot product attention than the basic single-stage sparse attention mechanisms. DuSA can still maintain low performance loss in some plug and play situations. DuSA can be used in both training and inference acceleration. After training, DuSA can even outperform VSA. DuSA achieves state-of-the-art performance in different benchmarks: long range arena, image classification, semantic segmentation, object detection, text to video generation, and long context understanding, and accelerates models of different sizes.

## Acknowledgements

This work is supported by the Hong Kong Innovation and Technology Commission (InnoHK Project CIMDA), the Institute of Digital Medicine, City University of Hong Kong (Projects 9229503 and 9610460), the National Natural Science Foundation of China (No. 12561095), and the Special Posts of Guizhou University (No. [2025]06).

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

## A1 Upper bounds analysis

The following derivations omit the normalization term of the softmax operation for convenience. To characterize the difference between $\boldsymbol{H}$ and $\boldsymbol{H}_1$, we set $\mathbb{I}_i = \{(i-1)b+1, (i-1)b+2, \ldots, ib\}$ with $i = 1, 2, \ldots, m/b$. Then the entries of $\boldsymbol{A}_1$ can also be defined as $\boldsymbol{A}_1(\mathbb{I}_i, \mathbb{I}_i) = \boldsymbol{A}(\mathbb{I}_i, \mathbb{I}_i)$ with $i = 1, 2, \ldots, m/b$, and $\boldsymbol{A}_1(\mathbb{I}_i, \mathbb{I}_{i'}) = \boldsymbol{0}_b$ with $i \neq i'$, where $\boldsymbol{0}_b \in \mathbb{R}^{b \times b}$ is the all-zeros matrix. Hence, we have

$$\|\boldsymbol{A}_1 - \boldsymbol{A}\|_F \leq \sum_{i,i'=1, i \neq i'}^{m/b} \|\boldsymbol{A}(\mathbb{I}_i, \mathbb{I}_{i'})\|_F,$$

which implies that

$$\|\boldsymbol{H}_1 - \boldsymbol{H}\|_F = \|\boldsymbol{A}_1 \boldsymbol{V} - \boldsymbol{A}\boldsymbol{V}\|_F \leq \sum_{i,i'=1, i \neq i'}^{m/b} \|\boldsymbol{A}(\mathbb{I}_i, \mathbb{I}_{i'})\|_F \|\boldsymbol{V}\|_F. \tag{A1}$$

For $\|\boldsymbol{A}' - \boldsymbol{A}\|_F$ and $\|\boldsymbol{A}'' - \boldsymbol{A}\|_F$, according to Remarks 1 and 2 we have

$$\|\boldsymbol{A}' - \boldsymbol{A}\|_F \leq \sum_{j,j'=1, j \neq j'}^{b} \|\boldsymbol{A}(\mathbb{J}_j, \mathbb{J}_{j'})\|_F; \tag{A2}$$

$$\|\boldsymbol{A}'' - \boldsymbol{A}\|_F \leq \sum_{k=2}^{b} \|\boldsymbol{A}''_k\|_F + \sum_{j,j'=1, j \neq j'}^{b} \|\boldsymbol{A}(\mathbb{J}_j, \mathbb{J}_{j'})\|_F$$

$$\leq \sum_{k=2}^{b} \sum_{j=1}^{b} \|\boldsymbol{A}(\mathbb{J}_j^{(k)}, \mathbb{J}_j^{(k)})\|_F + \sum_{j,j'=1, j \neq j'}^{b} \|\boldsymbol{A}(\mathbb{J}_j^{(1)}, \mathbb{J}_{j'}^{(1)})\|_F. \tag{A3}$$

*Remark* 3. Based on the above descriptions and Eqs. (6) and (7), the upper bound for $\|\boldsymbol{H}_2 - \boldsymbol{H}\|_F$ can be discussed in following two cases.

Case (6) corresponding to Eq. (6): an upper bound for $\|\boldsymbol{H}_2 - \boldsymbol{H}\|_F$ is deduced as

$$\|\boldsymbol{H}_2 - \boldsymbol{H}\|_F = \|\boldsymbol{A}'' \boldsymbol{A}_1 \boldsymbol{V} - \boldsymbol{A}\boldsymbol{V}\|_F = \|\boldsymbol{A}'' \boldsymbol{A}_1 \boldsymbol{V} - \boldsymbol{A}_1 \boldsymbol{V} + \boldsymbol{A}_1 \boldsymbol{V} - \boldsymbol{A}\boldsymbol{V}\|_F$$

$$\leq \|\boldsymbol{A}'' \boldsymbol{A}_1 \boldsymbol{V} - \boldsymbol{A}_1 \boldsymbol{V}\|_F + \|\boldsymbol{A}_1 \boldsymbol{V} - \boldsymbol{A}\boldsymbol{V}\|_F$$

$$\leq \|\boldsymbol{A}'' - \boldsymbol{I}_m\|_{\max} \|\boldsymbol{A}_1 \boldsymbol{V}\|_F + \sum_{i,i'=1, i \neq i'}^{m/b} \|\boldsymbol{A}(\mathbb{I}_i, \mathbb{I}_{i'})\|_F \|\boldsymbol{V}\|_F$$

$$\leq \|\boldsymbol{A}'' - \boldsymbol{I}_m\|_{\max} \|\boldsymbol{A}\boldsymbol{V}\|_F + (\|\boldsymbol{A}'' - \boldsymbol{I}_m\|_{\max} + 1) \sum_{i,i'=1, i \neq i'}^{m/b} \|\boldsymbol{A}(\mathbb{I}_i, \mathbb{I}_{i'})\|_F \|\boldsymbol{V}\|_F; \quad \text{(A4)}$$

Case (7) corresponding to Eq. (7): similar to Case (6), an upper bound for $\|\boldsymbol{H}_2 - \boldsymbol{H}\|_F$ is given as

$$\|\boldsymbol{H}_2 - \boldsymbol{H}\|_F \leq \|\boldsymbol{A}' - \boldsymbol{I}_m\|_{\max} \|\boldsymbol{A}\boldsymbol{V}\|_F$$

$$+ (\|\boldsymbol{A}' - \boldsymbol{I}_m\|_{\max} + 1) \sum_{i,i'=1, i \neq i'}^{m/b} \|\boldsymbol{A}(\mathbb{I}_i, \mathbb{I}_{i'})\|_F \|\boldsymbol{V}\|_F, \tag{A5}$$

where for any matrix $\boldsymbol{B} \in \mathbb{R}^{m \times n}$, the symbol $\|\boldsymbol{B}\|_{\max}$ is defined as

$$\|\boldsymbol{B}\|_{\max} = \max\{|b_{ij}| : i = 1, 2, \ldots, m; j = 1, 2, \ldots, n\}.$$

We now estimate two terms $\|\boldsymbol{A}'' - \boldsymbol{I}_m\|_{\max}$ and $\|\boldsymbol{A}' - \boldsymbol{I}_m\|_{\max}$ in Remark 3 based on Inequalities (A2) and (A3) as follows:

$$\|\boldsymbol{A}'' - \boldsymbol{I}_m\|_{\max} = \|\boldsymbol{A}'' - \boldsymbol{A} + \boldsymbol{A} - \boldsymbol{I}_m\|_{\max} \leq \|\boldsymbol{A}'' - \boldsymbol{A}\|_{\max} + \|\boldsymbol{A} - \boldsymbol{I}_m\|_{\max} \leq \|\boldsymbol{A}'' - \boldsymbol{A}\|_F$$

$$+ \|\boldsymbol{A} - \boldsymbol{I}_m\|_{\max} \leq \sum_{k=2}^{b} \sum_{j=1}^{b} \|\boldsymbol{A}(\mathbb{J}_j^{(k)}, \mathbb{J}_j^{(k)})\|_F + \sum_{j,j'=1, j \neq j'}^{b} \|\boldsymbol{A}(\mathbb{J}_j^{(1)}, \mathbb{J}_{j'}^{(1)})\|_F + \|\boldsymbol{A} - \boldsymbol{I}_m\|_{\max},$$

$$\tag{A6}$$

and

$$\|\boldsymbol{A}' - \boldsymbol{I}_m\|_{\max} \leq \sum_{j,j'=1,j\neq j'}^{b} \|\boldsymbol{A}(\mathbb{J}_j, \mathbb{J}_{j'})\|_F + \|\boldsymbol{A} - \boldsymbol{I}_m\|_{\max}. \tag{A7}$$

Therefore, the upper bound for $\|\boldsymbol{H}_2 - \boldsymbol{H}\|_F$ in Case (6) is obtained by combining Inequalities (A4) and (A6):

$$\|\boldsymbol{H}_2 - \boldsymbol{H}\|_F \leq \alpha\|\boldsymbol{AV}\|_F + \beta \sum_{i,i'=1,i\neq i'}^{m/b} \|\boldsymbol{A}(\mathbb{I}_i, \mathbb{I}_{i'})\|_F \|\boldsymbol{V}\|_F \tag{A8}$$

with

$$\alpha = \sum_{k=2}^{b} \sum_{j=1}^{b} \|\boldsymbol{A}(\mathbb{J}_j^{(k)}, \mathbb{J}_j^{(k)})\|_F + \sum_{j,j'=1,j\neq j'}^{b} \|\boldsymbol{A}(\mathbb{J}_j^{(1)}, \mathbb{J}_{j'}^{(1)})\|_F + \|\boldsymbol{A} - \boldsymbol{I}_m\|_{\max}, \quad \beta = \alpha + 1,$$

and the upper bound for $\|\boldsymbol{H}_2 - \boldsymbol{H}\|_F$ in Case (7) is obtained by combining Inequalities (A5) and (A7):

$$\|\boldsymbol{H}_2 - \boldsymbol{H}\|_F \leq \alpha'\|\boldsymbol{AV}\|_F + \beta' \sum_{i,i'=1,i\neq i'}^{m/b} \|\boldsymbol{A}(\mathbb{I}_i, \mathbb{I}_{i'})\|_F \|\boldsymbol{V}\|_F \tag{A9}$$

with

$$\alpha' = \sum_{j,j'=1,j\neq j'}^{b} \|\boldsymbol{A}(\mathbb{J}_j, \mathbb{J}_{j'})\|_F + \|\boldsymbol{A} - \boldsymbol{I}_m\|_{\max}, \quad \beta' = \alpha' + 1.$$

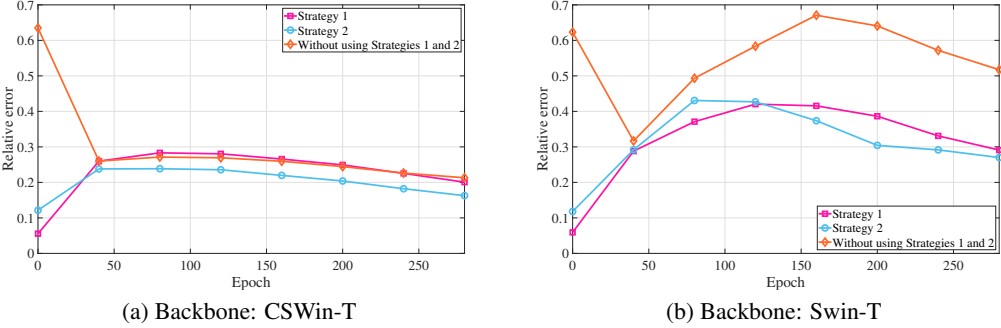

(a) Backbone: CSWin-T        (b) Backbone: Swin-T

Figure A1: The comparison of relative error of DuSA using Strategy 1 (its relative approximation error is: $\frac{\|\boldsymbol{H}_2 - \boldsymbol{H}\|_F}{\|\boldsymbol{H}\|_F}$ and the upper bound of the term $\|\boldsymbol{H}_2 - \boldsymbol{H}\|_F$ can be obtained using Inequality (A9)), DuSA using Strategy 2 (its relative approximation error is: $\frac{\|\boldsymbol{H}_2 - \boldsymbol{H}\|_F}{\|\boldsymbol{H}\|_F}$ and the upper bound of the term $\|\boldsymbol{H}_2 - \boldsymbol{H}\|_F$ can be obtained using Inequality (A8)), and DuSA without using Strategies 1 and 2 (its relative approximation error is: $\frac{\|\boldsymbol{H}_1 - \boldsymbol{H}\|_F}{\|\boldsymbol{H}\|_F}$ and the upper bound of the term $\|\boldsymbol{H}_1 - \boldsymbol{H}\|_F$ can be obtained using Inequality (A1)) in approximating VSA during the training of CSWin-T/Swin-T on ImageNet-1K. The approximation error shows a decreasing trend. Strategy 1 shows a lower error curve at the beginning of training which is consistent with the analysis of the relationship between the upper bound Inequality (A8) and the upper bound Inequality (A9). As the training process progresses, the actual approximation error of Strategy 2 becomes even smaller than the actual approximation error of Strategy 1 which is also consistent with more token information introduced by Strategy 2 in token mixing to enhance performance. Even the backbone is untrained, DuSA using Strategy 1 or 2 still provides significantly lower approximation error than DuSA without using Strategies 1 and 2 which further demonstrates that DuSA is more similar to VSA than simple sliding window based sparse attention.

*Remark* 4. For the term $\|\boldsymbol{H}_2 - \boldsymbol{H}\|_F$, its upper bound (A8) or (A9) may be relatively rough. In the future, a key issue is to obtain a tighter upper bound for $\|\boldsymbol{H}_2 - \boldsymbol{H}\|_F$ in Cases (6) and (7). Fig. A1 shows the comparison of the actual relative error of DuSA using blockify Strategy 1, DuSA using blockify Strategy 2, and DuSA without using blockify Strategies 1 and 2 in approximating VSA. Blockify Strategy 1 shows a lower error curve at the beginning of training which is consistent with the analysis of the relationship between the upper bound Inequality (A8) and the upper bound Inequality (A9). As the training process progresses, the actual approximation error of blockify Strategy 2 becomes even smaller than the actual approximation error of blockify Strategy 1 which is also consistent with more token information introduced by blockify Strategy 2 in token mixing to enhance performance. Even the backbone is untrained, DuSA using blockify Strategy 1 or 2 still provides significantly lower approximation error than DuSA without using blockify Strategies 1 and 2 which further demonstrates that DuSA is more similar to VSA than simple sliding window based sparse attention (DuSA without using blockify Strategies 1 and 2).

Table A1: ImageNet-1K classification comparison. Note: FPS was measured on 1 NVIDIA H20 using a batch size of 512 for tiny models and 256/32 for base models with a resolution of $224^2/384^2$ and mixed precision.

| Method | Resolution | Accuracy (%) | FPS (imgs/s) | # | FLOPs |
|---|---|---|---|---|---|
| | | | CSWin-B | | |
| Agent | $224^2$ \| $384^2$ | 84.7 \| 85.8 | 994 \| 276 | 73M | 14.49G \| 42.57G |
| EFFATT | $224^2$ \| $384^2$ | 84.4 \| 85.7 | 1059 \| 331 | 73M | 14.53G \| 42.69G |
| ELFATT | $224^2$ \| $384^2$ | 84.7 \| 85.8 | 1187 \| 355 | 73M | 14.46G \| 42.48G |
| FLatten | $224^2$ \| $384^2$ | 84.5 \| 85.5 | 864 \| 266 | 75M | 14.52G \| 42.67G |
| LOCAL | $224^2$ \| $384^2$ | 84.4 \| 85.5 | 1037 \| 323 | 73M | 14.39G \| 42.28G |
| VSA | $224^2$ \| $384^2$ | 84.7 \| 85.9 | 478/879 (FA2) \| 82/201 (FA2) | 73M | 22.33G/14.39G (FA2) \| 110.89G/42.28G (FA2) |
| DuSA | $224^2$ \| $384^2$ | 84.7 \| 85.9 | 1203 \| 356 | 73M | 14.39G \| 42.28G |
| | | | CSWin-T | | |
| Agent | $224^2$ | 83.1 | 2425 | 20M | 4.14G |
| EFFATT | $224^2$ | 82.6 | 2526 | 20M | 4.17G |
| ELFATT | $224^2$ | 83.1 | 2856 | 20M | 4.13G |
| FLatten | $224^2$ | 83.1 | 2025 | 21M | 4.16G |
| LOCAL | $224^2$ | 82.7 | 2519 | 20M | 4.09G |
| VSA | $224^2$ | 83.1 | 1303/2210 (FA2) | 20M | 7.60G/4.09G (FA2) |
| DuSA | $224^2$ | 83.2 | 2908 | 20M | 4.09G |
| | | | Swin-B | | |
| Agent | $224^2$ \| $384^2$ | 84.0 \| 84.9 | 1367 \| 372 | 88M | 15.44G \| 46.34G |
| EFFATT | $224^2$ \| $384^2$ | 84.2 \| 85.3 | 1536 \| 481 | 88M | 15.33G \| 45.04G |
| ELFATT | $224^2$ \| $384^2$ | 84.5 \| 85.5 | 1497 \| 457 | 91M | 15.68G \| 46.08G |
| FLatten | $224^2$ \| $384^2$ | 83.8 \| 85.0 | 1226 \| 353 | 89M | 15.46G \| 46.49G |
| LOCAL | $224^2$ \| $384^2$ | 83.5 \| 84.5 | 1351 \| 357 | 88M | 15.47G \| 47.19G |
| VSA | $224^2$ \| $384^2$ | 84.2 \| 85.3 | 743/1325 (FA2) \| 134/313 (FA2) | 88M | 21.57G/15.19G (FA2) \| 99.76G/44.64G (FA2) |
| DuSA | $224^2$ \| $384^2$ | 84.5 \| 85.5 | 1568 \| 485 | 91M | 15.61G \| 45.88G |
| | | | Swin-T | | |
| Agent | $224^2$ | 82.6 | 2847 | 29M | 4.53G |
| EFFATT | $224^2$ | 82.1 | 3282 | 28M | 4.45G |
| ELFATT | $224^2$ | 82.7 | 3159 | 30M | 4.67G |
| FLatten | $224^2$ | 82.1 | 2502 | 29M | 4.50G |
| LOCAL | $224^2$ | 81.4 | 2943 | 28M | 4.51G |
| VSA | $224^2$ | 82.4 | 1269/2571 (FA2) | 28M | 8.81G/4.38G (FA2) |
| DuSA | $224^2$ | 82.7 | 3154 | 30M | 4.63G |
| | | | Others | | |
| ConvNeXt-T | $224^2$ | 82.1 | 3911 | 29M | 4.47G |
| VMamba-T | $224^2$ | 82.5 | 1837 | 30M | 4.84G |

## A2 Complexity analysis

For VSA, the nature of the scaled-dot product attention calculation leads to the complexity of $O(m^2 \times c)$. For blockify Strategy 1, the complexity of DuSA is determined by Eqs. (3) and (4). In Eq. (3), the complexity of the computation of the attention matrix $\boldsymbol{A}_1$ is $O(m \times b \times c)$ and the complexity of the token mixing to obtain $\boldsymbol{H}'$ is also $O(m \times b \times c)$. Hence, the total complexity of Eq. (3) is $O(m \times b \times c)$. Similarly, for the complexity of Eq. (4), its complexity is $O((m^2/b) \times c)$. The total complexity of DuSA using blockify Strategy 1 is $O(\frac{m \times b^2 + m^2}{b} \times c)$, and it can achieve the

minimum $O(m \times \sqrt{m} \times c)$ when $b = \sqrt{m}$. For blockify Strategy 2, compared to blockify Strategy 1, it only has one more attention score aggregation process, as shown in Eq. (5) of which the complexity is $O(m^2/b) \ll O((m^2/b) \times c)$. Hence, the total complexity of DuSA using blockify Strategy 2 is also $O(\frac{m \times b^2 + m^2}{b} \times c)$ and its minimum is also $O(m \times \sqrt{m} \times c)$ when $b = \sqrt{m}$. If $m \gg c$, the complexity of DuSA is $O(m \times \sqrt{m})$, which is less than $O(m^2)$ of VSA. Our method is designed for acceleration when $m \gg c$. The larger $m$ is than $c$, the more obvious the acceleration effect is.

Table A2: The effect of different block sizes at each level (L1-L4) on the performance of DuSA and some plug and play (PP) examples using DuSA to replace VSA directly without training in the ImageNet-1K classification task. Note: FPS was measured on 1 NVIDIA H20 using a batch size of 512 and mixed precision. "PT" denotes the models pre-trained from scratch. CSWin-T-VSA here is the same as CSWin-T-VSA in Table A1 while Swin-T-VSA here uses the enhanced version from [18] which replaces the concatenation operations for partition/merging with convolutions to enhance performance, hence it is a little different from Swin-T-VSA in Table A1.

| Block size of each level | | | | Resolution | Accuracy (PT/PP) | FPS (imgs/s) | # | FLOPs |
| L1 | L2 | L3 | L4 | | | | | |
|---|---|---|---|---|---|---|---|---|
| | | | | | CSWin-T | | | |
| 49 | 49 | 196 | 49 | $224^2$ | 82.7/78.8 | 2989 | 20M | 4.09G |
| 196 | 196 | 196 | 49 | $224^2$ | 83.2/82.0 | 2908 | 20M | 4.09G |
| 784 | 784 | 196 | 49 | $224^2$ | 83.2/83.1 | 2860 | 20M | 4.09G |
| VSA (full sequence length) | | | | $224^2$ | 83.1/ — | 1303/2210 (FA2) | 20M | 7.60G/4.09G (FA2) |
| 3136 | 784 | 196 | 49 | | | | | |
| | | | | | Swin-T | | | |
| 49 | 49 | 196 | 49 | $224^2$ | 82.5/73.1 | 3223 | 30M | 4.63G |
| 196 | 196 | 196 | 49 | $224^2$ | 82.7/80.3 | 3154 | 30M | 4.63G |
| 784 | 784 | 196 | 49 | $224^2$ | 82.8/82.4 | 2958 | 30M | 4.63G |
| VSA (full sequence length) | | | | $224^2$ | 82.7/ — | 1188/2257 (FA2) | 30M | 9.06G/4.63G (FA2) |
| 3136 | 784 | 196 | 49 | | | | | |

## A3 The effect of different block sizes and plug and play examples

Table A2 shows the effect of different block sizes at each level (L1-L4) on the performance of DuSA and some plug and play (PP) examples using DuSA to replace VSA directly without training in the ImageNet-1K classification task. The sequence lengths of VSA in 4 levels of CSWin-T/Swin-T are $[3136, 784, 196, 49]$. The sequence lengths of the last two levels are too short to affect ViT speed [7, 16–18]. Hence, we only modified the block sizes of the first two levels. As the block sizes increase, the inference speed is reduced and the accuracy is improved. Even using the 75% sparsity ratio of the first level, DuSA still offers a $1.3\times$ speedup over FA2 in Table A2 and outperforms VSA. Also, as Table A2 shows, when using the 75%-94% sparsity ratios of the first two levels, DuSA can almost maintain 97%+ accuracy in a plug and play application. The current version of DuSA is a static sparse attention mechanism, which cannot preserve the accuracy of VSA without loss in a plug and play application. It is designed for accelerating both the training and inference processes. After training, it can match or even outperform VSA. Inspired by XAttention, we have provided a training-free dynamic sparse attention version of DuSA for accelerating the prefilling stage of large language models (LLMs). The dynamic version of DuSA can match the performance of VSA in plug and play applications and provides a significant acceleration effect. Details are available in Section A6.

## A4 The use of DuSA in simple and general architectures

Table A3 shows the performance comparison of vanilla ViT architectures and the versions using DuSA to replace VSA. The sequence length $m$ of the vanilla ViT architecture is relatively short (for the input resolution of $224^2/384^2$ using a constant input patch size of 16, $m = 197/577$ with a constant embedding dimension $c = 64$), which limits the acceleration effect of DuSA and cannot completely show the superiority of DuSA for acceleration. Because DuSA is designed for acceleration when $m \gg c$.

Table A3: The performance comparison of vanilla ViT architectures and the versions using DuSA to replace VSA. All models are trained from scratch on ImageNet-1K. Block size $b$ is $14/24$ for tiny(T)/large(L) models. Padding is used to make the sequence length divisible by $b$. FPS and the inference memory are obtained on 1 NVIDIA H20 using a batch size of 512/32 for the T/L models.

| Method | Resolution | Accuracy (%) | FPS (imgs/s) | # | FLOPs | Inference memory (GB) |
|---|---|---|---|---|---|---|
| ViT-L/16-VSA | $384^2$ | 76.5 | 179/272 (FA2) | 307M | 191.21G/174.85G (FA2) | 3.7/**2.9** (FA2) |
| ViT-L/16-DuSA | $384^2$ | **77.2** | 230 | 307M | 174.54G | 3.0 |
| ViT-T/16-VSA | $224^2$ | 72.6 | 10644/15350 (FA2) | 6M | 1.26G/1.08G (FA2) | 2.2/**2.0** (FA2) |
| ViT-T/16-DuSA | $224^2$ | **72.9** | 10718 | 6M | 1.07G | **2.0** |

## A5 Memory consumption patterns of DuSA

Tables A4 and A5 show the memory comparison results. Compared to ELFATT and FA2, DuSA still has the overall lowest memory usage and its memory usage is significantly lower than that of VSA.

Table A4: Peak memory usage comparison on ImageNet-1K using 1 NVIDIA H20 (96GB). The batch size used is 32 in both inference and training. "OOM" denotes out of memory (MEM).

| Method | Resolution | Inference MEM (GB) | Training MEM (GB) |
|---|---|---|---|
| CSWin-B-ELFATT | $384^2$ | 2.5 | 18.8 |
| CSWin-B-VSA | $384^2$ | 62.3/2.4 (FA2) | OOM/**17.9** (FA2) |
| CSWin-B-DuSA | $384^2$ | **2.3** | 21.0 |

Table A5: Peak memory usage comparison on ADE20K and MS COCO 2017 using 1 NVIDIA H20 (96GB). The batch size used is 1 for inference and 2 for training. "OOM" denotes out of memory.

| Method | ADE20K inference MEM (GB) | COCO inference MEM (GB) | ADE20K training MEM (GB) | COCO training MEM (GB) |
|---|---|---|---|---|
| | | Swin-T | | |
| ELFATT | 2.8 | 1.6 | 5.6 | 8.2 |
| VSA | 49.2/2.8 (FA2) | 71.9/1.5 (FA2) | 20.1/**5.5** (FA2) | OOM/8.4 (FA2) |
| DuSA | **2.5** | **1.4** | **5.5** | **8.1** |
| | | CSWin-T | | |
| ELFATT | **2.7** | **1.4** | 5.5 | 8.8 |
| VSA | 33.2/**2.7** (FA2) | 48.2/**1.4** (FA2) | 16.7/**5.4** (FA2) | OOM/8.8 (FA2) |
| DuSA | **2.7** | **1.4** | **5.4** | **8.7** |

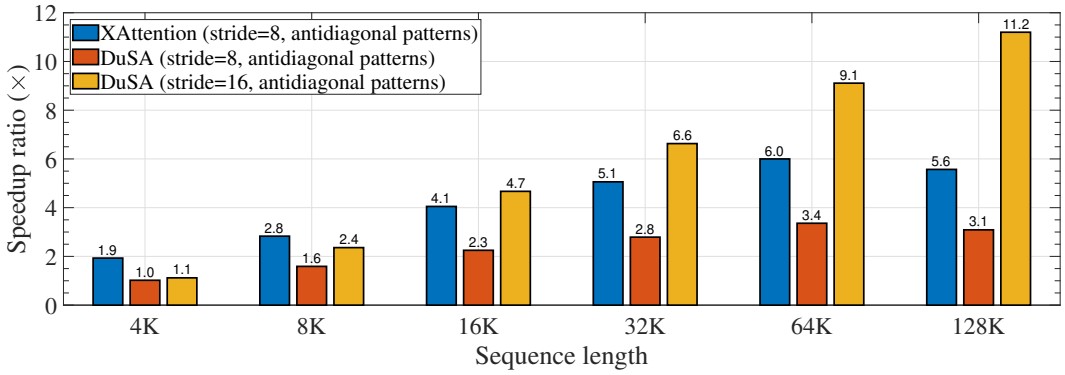

Figure A2: Speedup relative to FlashAttention of FlashInfer obtained by different methods across different sequence lengths using Llama-3.1-8B-Instruct on 1 NVIDIA H20.

## A6 The application of DuSA in accelerating the prefilling stage of LLMs

Based on XAttention, we introduce DuSA (Strategy 2: Rolling indices-based attention scores fusion) to search antidiagonal/diagonal patterns of attention matrices to filter important or unimportant blocks for further dynamic sparse attention performed within important blocks only. As Fig. A2 and Table

A6 show, DuSA can use a larger stride (larger sparsity) to achieve similar or higher performance than XAttention, further accelerating FlashAttention of FlashInfer. By introducing antidiagonal patterns of XAttention, the performance of DuSA can be further improved.

Table A6: The performance comparison of different attention methods on LongBench using the Llama-3.1-8B-Instruct model. Note: Higher scores mean better performance.

| NarrativeQA | Qasper | MultiFieldQA-en | MultiFieldQA-zh | HotpotQA | 2WikiMultihopQA | MuSiQue | DuReader | GovReport | QMSum | MultiNews | VCSUM | TREC | TriviaQA | SAMSum | LSHT | PassageCount | PassageRetrieval-en | PassageRetrieval-zh | LCC | RepoBench-P | Average |
|---|---|---|---|---|---|---|---|---|---|---|---|---|---|---|---|---|---|---|---|---|---|
| VSA (FlashAttention of FlashInfer) | | | | | | | | | | | | | | | | | | | | | |
| **31.44** | 25.07 | **29.40** | **61.68** | 16.89 | 17.00 | 11.79 | 34.93 | 34.22 | 23.25 | 26.69 | 15.91 | 72.50 | 91.65 | 43.74 | 46.00 | 5.95 | **98.20** | **77.11** | 52.19 | 49.14 | **41.18** |
| XAttention (stride=8, antidiagonal patterns) | | | | | | | | | | | | | | | | | | | | | |
| 30.48 | 26.04 | 29.28 | 61.67 | 17.33 | 16.34 | 11.88 | 34.64 | 34.60 | 23.24 | 27.08 | 16.11 | 71.50 | 90.97 | **44.13** | 46.50 | 5.23 | 88.68 | 74.40 | 53.23 | **50.94** | 40.68 |
| DuSA (stride=8, antidiagonal patterns) | | | | | | | | | | | | | | | | | | | | | |
| 30.81 | 24.28 | 28.88 | 58.28 | **18.09** | 16.03 | 10.21 | 35.40 | **34.68** | 23.21 | 26.92 | 15.77 | 72.50 | **92.15** | 43.27 | **47.00** | **7.25** | 96.84 | 76.67 | 52.28 | 48.97 | 40.92 |
| DuSA (stride=8, diagonal patterns) | | | | | | | | | | | | | | | | | | | | | |
| 31.34 | 25.96 | 28.78 | 61.65 | 16.92 | 16.35 | 11.86 | 35.60 | 34.54 | **23.43** | **27.17** | **16.21** | **73.00** | 90.62 | 43.59 | 46.50 | 6.38 | 89.26 | 75.06 | 53.24 | 50.16 | 40.84 |
| DuSA (stride=16, antidiagonal patterns) | | | | | | | | | | | | | | | | | | | | | |
| 30.21 | **27.57** | 29.00 | 60.99 | 17.34 | **17.02** | **12.04** | **36.13** | 34.59 | 23.18 | 26.96 | 15.73 | 71.50 | 90.90 | 43.67 | 46.00 | 6.39 | 97.76 | 73.76 | **53.36** | 49.56 | 41.13 |

## A7 Broader impacts

DuSA provides an efficient attention mechanism to accelerate training and inference of long-context transformers, and it is not hardware-aware, which will benefit academia and startups to reduce deployment costs using medium-performance GPUs and unlock cross-disciplinary breakthroughs. DuSA can be used in advanced hierarchical sparse attention mechanisms to improve their performance and speed as a novel alternative for basic sparse attention. Although the enhanced efficiency of long-context transformers may increase risks such as misuse in generating fake/harmful contents, the benefits brought by efficient long-context transformers outweigh the risks in general.

## A8 Limitations

As Remark 4 shows, for the term $\|\boldsymbol{H}_2 - \boldsymbol{H}\|_F$, its upper bound (Inequality (A8) or (A9)) provides a worst-case guarantee and may be relatively rough: In the current version, the upper bound (Inequality (A8) or (A9)) for Strategy 1 or 2 is larger than the upper bound (Inequality (A1)) for simple block sparse attention. However, as shown in Fig. A1, the actual approximation error of DuSA (Strategy 1) or DuSA (Strategy 2) is usually much lower than that of DuSA without using Strategies 1 and 2. In the future, a key issue is to obtain a tighter upper bound for Strategy 1 or 2. In addition, the acceleration effect of DuSA may be limited when $m$ is not significantly larger than $c$.

