# OpenReview forum: "DuSA: Fast and Accurate Dual-Stage Sparse Attention Mechanism Accelerating Both Training and Inference"
_NeurIPS.cc/2025/Conference — NeurIPS 2025 poster_

### Official Review · Reviewer_Ah1n · 2025-07-01

**Clarity:** 3
**Significance:** 2
**Originality:** 3
**Rating:** 3
**Confidence:** 5

**Summary:**

DuSA introduces a novel Dual-Stage Sparse Attention method to accelerate both the training and inference of vision transformers. The main idea is to split the attention computation into two stages. Both stages implement sparse attention: the first stage is designed to capture local inductive biases within individual blocks, while the second stage aggregates cross-block information. Since both stages can work with the existing attention acceleration techniques, and sparse attention has low computational complexity, DuSA achieves a significant speedup over the vanilla scaled-dot product attention (VSA). Experiments on different benchmarks demonstrate the effectiveness of the proposed DuSA method.

**Questions:**

- There's been a lot of recent works on sparse attention for LLMs. Can DuSA be applied to LLMs — for both prefill and decoding?
- DuSA is similar to a recent paper XAttention [1], they both use block sparsity and diagonal sparse patterns. From the results of XAttention, antidiagonal patterns outperform diagonal ones, which is leveraged in DuSA. Could you compare DuSA and XAttention in more detail?

[1] Ruyi Xu, et al. XAttention: Block Sparse Attention with Antidiagonal Scoring. arXiv:2503.16428.

**Ethical Concerns:**

["NO or VERY MINOR ethics concerns only"]

**Limitations:**

yes

**Paper Formatting Concerns:**

No formatting concerns.

**Quality:**

2

**Strengths And Weaknesses:**

- Strengths
	- The proposed method is compatible with existing efficient attention implementations, allowing it to be easily integrated into the existing vision transformer models to get better performance. The experimental results also show the real speedup, which makes the method solid and practical.
- Weakness
	- This paper spends a considerable amount of space analyzing the upper bound of the approximation error, but this bound is not used to guide the design of sparse attention. Additionally, as the author noted, this upper bound is relatively rough and does not seem helpful for sparse attention analysis and design right now (though that could change later). Maybe it’d be better to move this section to the appendix.
	- Many discussions and ablation studies are in the appendix. It'd be better to have them in the "Experiments and Results" session to help readers understand.
	- In the abstract, the paper claims that DuSA maintains low performance loss in plug-and-play scenarios. But this topic is only discussed in Section A2, with evaluations limited to ImageNet. The results show a significant drop in accuracy.

---

> ### Author Rebuttal · Authors · 2025-07-30
>
> **We would like to thank the reviewer for the constructive comments provided to help us improve the quality of the paper. We have addressed the issues raised in the comments point-by-point as follows:**
>
> ---
>
> **Response to W1 and W2:** We will move the section about upper bound analysis of the approximation error to the appendix and move more experimental discussions and ablation studies to the main body in the next version.
>
> ---
>
> **Response to W3:** The current version of DuSA is a static sparse attention mechanism, which cannot preserve the accuracy of VSA without loss in a plug-and-play application. It is designed for accelerating both the training and inference processes. After training, it can match or even outperform VSA. Inspired by XAttention [1], which is an excellent training-free dynamic sparse attention SOTA for accelerating the prefilling stage of LLMs, we have provided a dynamic sparse attention version of DuSA. The dynamic version of DuSA can match the performance of VSA in plug-and-play applications and provides a significant acceleration effect. Details are available in **Response to Q1**.
>
> ---
>
> **Response to Q1:** The answer is yes. DuSA can be applied to accelerate the prefilling stage. Based on XAttention [1], we introduce DuSA2 (Strategy 2: Rolling indices-based attention scores fusion) to search antidiagonal/diagonal patterns of attention matrices to filter important or unimportant blocks for further dynamic sparse attention performed within important blocks only. As the following **Table R4-1** shows, compared to XAttention, the performance of DuSA2 is closer to that of full attention. As **Tables R4-1** and **R4-2** show, DuSA2 can use a larger stride (larger sparsity) to achieve similar or higher performance than XAttention, further accelerating full attention. By introducing antidiagonal patterns of XAttention, the performance of DuSA2 can be further improved.
>
> ## Table R4-1. The performance comparison of different attention mechanisms on LongBench using the Llama-3.1-8B-Instruct model. Note: Higher scores mean better performance.
> |Method|Full (FlashInfer’s implementation of FlashAttention)|XAttention (stride=8) antidiagonal patterns|DuSA2 (stride=8) antidiagonal patterns|DuSA2 (stride=8) diagonal patterns|DuSA2 (stride=16) antidiagonal patterns|
> |-|-|-|-|-|-|
> |NarrativeQA|31.44|30.48|30.81|31.34|30.21|
> |Qasper|25.07|26.04|24.28|25.96|**27.57**|
> |MultiFieldQA-en|**29.40**|29.28|28.88|28.78|29.00|
> |MultiFieldQA-zh|**61.68**|61.67|58.28|61.65|60.99|
> |HotpotQA|16.89|17.33|**18.09**|16.92|17.34|
> |2WikiMultihopQA|17.00|16.34|16.03|16.35|**17.02**|
> |MuSiQue|11.79|11.88|10.21|11.86|**12.04**|
> |DuReader|34.93|34.64|35.40|35.60|**36.13**|
> |GovReport|34.22|34.60|**34.68**|34.54|34.59|
> |QMSum|23.25|23.24|23.21|**23.43**|23.18|
> |MultiNews|26.69|27.08|26.92|**27.17**|26.96|
> |VCSUM|15.91|16.11|15.77|**16.21**|15.73|
> |TREC|72.50|71.50|72.50|**73.00**|71.50|
> |TriviaQA|91.65|90.97|**92.15**|90.62|90.90|
> |SAMSum|43.74|**44.13**|43.27|43.59|43.67|
> |LSHT|46.00|46.50|**47.00**|46.50|46.00|
> |PassageCount|5.95|5.23|**7.25**|6.38|6.39|
> |PassageRetrieval-en|**98.20**|88.68|96.84|89.26|97.76|
> |PassageRetrieval-zh|**77.11**|74.40|76.67|75.06|73.76|
> |LCC|52.19|53.23|52.28|53.24|**53.36**|
> |RepoBench-P|49.14|**50.94**|48.97|50.16|49.56|
> |Average|**41.18**|40.68|40.92|40.84|41.13|
>
> ## Table R4-2. Speedup relative to FlashAttention (FlashInfer’s implementation) obtained by different attention mechanisms across different sequence lengths using Llama-3.1-8B-Instruct on a single H20 (96GB) GPU.
> |Sequence Length (K)|DuSA2 (stride=8) Speedup|DuSA2 (stride=16) Speedup|XAttention (stride=8) Speedup|
> |-|-|-|-|
> |4|1.02|1.12|**1.93**|
> |8|1.59|2.36|**2.83**|
> |16|2.25|**4.67**|4.05|
> |32|2.79|**6.63**|5.06|
> |64|3.36|**9.11**|6.00|
> |128|3.09|**11.20** |5.57|
>
> ---
>
> **Response to Q2:** XAttention uses the sum of antidiagonal/diagonal values of the attention matrix as the metric for ranking block importance. While in stage 2, DuSA uses the diagonal values of the attention matrix for block-level mixture to introduce long-range dependencies from other diagonal blocks from stage 1. We take the version of XAttention that uses diagonal patterns for comparison.  For the input $\textbf{Q}$ and $\textbf{K}$ with a size of $(B, H, M, C)$, where $B$ is the batch size, $H$ is the number of heads, $M$ is the sequence length, and $C$ is the embedding dimension of each head, the version of XAttention using diagonal patterns reshapes the $\textbf{Q}$ and $\textbf{K}$ to a shape like $(B, H, M//stride, stride\times C)$ and performs attention computation using softmax on the final two dimensions to obtain diagonal values of the attention matrices. While DuSA1 (DuSA using Strategy 1) reshapes the $\textbf{Q}$ and $\textbf{K}$ to a shape of $(B, H, M//stride, stride, C)$ and transposes the third and fourth dimensions to a shape like $(B, H, stride, M//stride, C)$. DuSA1 also performs attention computation using softmax on the final two dimensions to obtain diagonal values of the attention matrices (*For satisfying the input requirements of some memory-efficient methods, $(B, H, stride, M//stride, C)$ may be further reshaped to $(B, H\times stride, M//stride, C)$. The softmax operation is still performed on the final two dimensions, hence they give the same results.*). The difference between DuSA1 and XAttention is that DuSA1 introduces a multi-stride computation to obtain diagonal values of the attention matrices. And DuSA2 (DuSA using Strategy 2) further sums the stride dimension to obtain $\textbf{A}_2$ as **Figure 3 of the original manuscript** shows.
>
> ---
>
> **References:**
>
> [1] Xu, R., Xiao, G., Huang, H., Guo, J., & Han, S. (2025). Xattention: Block sparse attention with antidiagonal scoring. *arXiv preprint arXiv:2503.16428*.

---

> > ### Comment · Reviewer_Ah1n · 2025-08-06
> >
> > Thanks for your detailed response. Your "Response to W1 and W2" comments strengthen the basis of your work and I hope you can fix them in the paper. I don't have any more questions for now and plan to keep the score.

---

> > > ### Author Response · Authors · 2025-08-07
> > > **Thank You for Your Response**
> > >
> > > Thank you for your response! We will fix W1 and W2 in the future version.
> > >
> > > Thank you again for your time and effort in reviewing our paper!
> > >
> > > ---
> > >
> > > Best regards,
> > >
> > > Authors

---

> ### Author Response · Authors · 2025-08-05
> **Looking Forward to Your Response**
>
> We would greatly appreciate it if you could tell us whether our rebuttal addresses your concerns. If there are no remaining concerns, we would greatly appreciate it if you could consider updating your evaluation of our paper. We welcome any further questions you may have about our work.
>
> We are looking forward to your response.
>
> Thank you for your time and effort in reviewing our paper!
>
> ---
>
> Best regards,
>
> Authors

---

### Official Review · Reviewer_iq85 · 2025-07-02

**Clarity:** 2
**Significance:** 3
**Originality:** 3
**Rating:** 4
**Confidence:** 1

**Summary:**

This paper proposes DuSA, which is a Dual-Stage Sparse Attention (DuSA) mechanism to accelerate the attention part of the vision transformers. A theoretical upper bound using DuSA to approximate VSA is derived.  Experiments demonstrated that the proposed DuSA has a lower approximation error than the single-stage design, despite the higher theoretical upper bound. DuSA can be used in both training and inference acceleration.

**Questions:**

1. Could the authors provide more explanation on larger theoretical upper bound but experimentally better results?
2. Any insights of the proposed techniques on general transformer architectures? Or is there any thing unique making the proposed techniques especially effective for Vits?

**Ethical Concerns:**

["NO or VERY MINOR ethics concerns only"]

**Final Justification:**

1. The performance is good; the additional stats by the authors further verify the savings memory usage
2. Still, all the tricks being put together seems to be a little too empirical and lacks enough analysis

**Limitations:**

yes

**Paper Formatting Concerns:**

No formatting issues

**Quality:**

3

**Strengths And Weaknesses:**

Strengths:

1. Theoretical derivation and analysis of the proposed attention mechanism
2. The plug and play nature of the technique at both training and inference
3. Overall good results


Weakness:

1. More insights on the experimentally shown good results given the slightly weaker theoretical analysis would be better
2. More analysis on memory consumption patterns during training would be more insightful
3. The plug-and-play's accuracy drops might be a little too large

---

> ### Author Rebuttal · Authors · 2025-07-30
>
> **We would like to thank the reviewer for the constructive comments provided to help us improve the quality of the paper. We have addressed the issues raised in the comments point-by-point as follows:**
>
> ---
>
> **Response to W1 and Q1:** The theoretical upper bound is used to provide the worst-case guarantee. A larger bound represents a larger confidence interval, and it may be larger than its real approximation error. Hence, as we mentioned in the limitation, a tighter bound is needed that can more accurately represent the real approximation error.
>
> ---
>
> **Response to W2:** **Tables R3-1** and **R3-2** show the memory comparison results. Even compared to SOTAs (ELFATT and FlashAttention-2 (FA2)), DuSA2 (DuSA using Strategy 2) still has the lowest memory usage, and its memory usage is significantly lower than that of VSA.
>
> ## Table R3-1. Peak memory usage comparison on ImageNet-1K using a single H20 (96GB) GPU. The batch size used is 32 for both inference and training. OOM denotes out of memory.
> |Methods|Resolution|Inference Memory (GB)|Training Memory (GB)|
> |-|-|-|-|
> |CSWin-B-DuSA2|$384^2$|**2.3**|21.0|
> |CSWin-B-ELFATT|$384^2$|2.5|18.8|
> |CSWin-B-VSA|$384^2$|62.3/2.4 (FA2)|OOM/**17.9 (FA2)**|
>
> ## Table R3-2. Peak memory usage comparison on ADE20K and COCO using a single H20 (96GB) GPU. The batch size used is 1 for inference and 2 for training. OOM denotes out of memory.
> |Methods|ADE20K (Inference Memory GB)|COCO (Inference Memory GB)|ADE20K (Training Memory GB)|COCO (Training Memory GB)|
> |-|-|-|-|-|
> |Swin-T-DuSA2|**2.5**|**1.4**|**5.5**|**8.1**|
> |Swin-T- ELFATT|2.8|1.6|5.6|8.2|
> |Swin-T-VSA|49.2/2.8 (FA2)|71.9/1.5 (FA2)|20.1/**5.5 (FA2)**|OOM/8.4 (FA2)|
> |CSWin-T-DuSA2|**2.7**|**1.4**|**5.4**|**8.7**|
> |CSWin-T-ELFATT|**2.7**|**1.4**|5.5|8.8|
> |CSWin-T-VSA|33.2/**2.7 (FA2)**|48.2/**1.4 (FA2)**|16.7/**5.4 (FA2)**|OOM/8.8 (FA2)|
>
> ---
>
> **Response to W3:** The current version of DuSA is a static sparse attention mechanism, which cannot preserve the accuracy of VSA without loss in a plug-and-play application. It is designed for accelerating both the training and inference processes. After training, it can match or even outperform VSA. And we have provided a dynamic sparse attention version, which is training-free and maintains higher performance. The dynamic version of DuSA can match the performance of VSA in a plug-and-play application and provides a significant acceleration effect. We introduce DuSA2 (Strategy 2: Rolling indices-based attention scores fusion) to search antidiagonal or diagonal patterns of attention matrices to filter important or unimportant blocks for further dynamic sparse attention only performed within important blocks. As the following **Table R3-3** shows, compared to XAttention [1], the performance of DuSA2 is closer to that of full attention. As **Tables R3-3** and **R3-4** show, DuSA2 can use a larger stride (larger sparsity) to achieve similar or higher performance than XAttention, further accelerating full attention. By introducing antidiagonal patterns of XAttention, the performance of DuSA2 can be further improved.
>
> ## Table R3-3. The performance comparison of different attention mechanisms on LongBench using the Llama-3.1-8B-Instruct model. Note: Higher scores mean better performance.
> |Method|Full (FlashInfer’s implementation of FlashAttention)|XAttention (stride=8) antidiagonal patterns|DuSA2 (stride=8) antidiagonal patterns|DuSA2 (stride=8) diagonal patterns|DuSA2 (stride=16) antidiagonal patterns|
> |-|-|-|-|-|-|
> |NarrativeQA|31.44|30.48|30.81|31.34|30.21|
> |Qasper|25.07|26.04|24.28|25.96|**27.57**|
> |MultiFieldQA-en|**29.40**|29.28|28.88|28.78|29.00|
> |MultiFieldQA-zh|**61.68**|61.67|58.28|61.65|60.99|
> |HotpotQA|16.89|17.33|**18.09**|16.92|17.34|
> |2WikiMultihopQA|17.00|16.34|16.03|16.35|**17.02**|
> |MuSiQue|11.79|11.88|10.21|11.86|**12.04**|
> |DuReader|34.93|34.64|35.40|35.60|**36.13**|
> |GovReport|34.22|34.60|**34.68**|34.54|34.59|
> |QMSum|23.25|23.24|23.21|**23.43**|23.18|
> |MultiNews|26.69|27.08|26.92|**27.17**|26.96|
> |VCSUM|15.91|16.11|15.77|**16.21**|15.73|
> |TREC|72.50|71.50|72.50|**73.00**|71.50|
> |TriviaQA|91.65|90.97|**92.15**|90.62|90.90|
> |SAMSum|43.74|**44.13**|43.27|43.59|43.67|
> |LSHT|46.00|46.50|**47.00**|46.50|46.00|
> |PassageCount|5.95|5.23|**7.25**|6.38|6.39|
> |PassageRetrieval-en|**98.20**|88.68|96.84|89.26|97.76|
> |PassageRetrieval-zh|**77.11**|74.40|76.67|75.06|73.76|
> |LCC|52.19|53.23|52.28|53.24|**53.36**|
> |RepoBench-P|49.14|**50.94**|48.97|50.16|49.56|
> |Average|**41.18**|40.68|40.92|40.84|41.13|
>
> ## Table R3-4. Speedup relative to FlashAttention (FlashInfer’s implementation) obtained by different attention mechanisms across different sequence lengths using Llama-3.1-8B-Instruct on a single H20 (96GB) GPU.
> |Sequence Length (K)|DuSA2 (stride=8) antidiagonal patterns Speedup|DuSA2 (stride=16) antidiagonal patterns Speedup|XAttention antidiagonal patterns (stride=8) Speedup|
> |-|-|-|-|
> |4|1.02|1.12|**1.93**|
> |8|1.59|2.36|**2.83**|
> |16|2.25|**4.67**|4.05|
> |32|2.79|**6.63**|5.06|
> |64|3.36|**9.11**|6.00|
> |128|3.09|**11.20** |5.57|
>
> ---
>
> **Response to Q2:** We have applied DuSA2 (DuSA using Strategy 2) to the simple vanilla ViT architectures (ViT-T/16 and ViT-L/16). **Table R3-5** shows the performance comparison of vanilla ViT architectures and the versions using DuSA2 to replace VSA. The maximum sequence length of the vanilla ViT architecture is relatively short. For the resolution $224^2$, its maximum sequence length is only $196+1$, and its four levels have sequence lengths: $197$-$197$-$197$-$197$ with a constant embedding dimension $c=64$ for each head. Even for the resolution $384^2$, its maximum sequence length is only $576+1$, and its four levels have sequence lengths: $577$-$577$-$577$-$577$ with a constant embedding dimension $c=64$ for each head. While the maximum sequence length of Swin and CSWin is $3136$/$9216$ for the resolution of $224^2$/$384^2$, and their four levels have sequence lengths: $3136$-$784$-$196$-$49$/$9216$-$2304$-$576$-$144$ for the resolution of $224^2$/$384^2$ with a constant embedding dimension $c=32$. As the ablation study of **Table A2 of the original manuscript** shows, if the sequence length $m$ is not significantly larger than the head embedding dimension $c$, the acceleration effect will be limited, which cannot completely show the superiority of our method for acceleration. Although the acceleration effect of DuSA on the ViT series is limited due to the short sequence length, DuSA achieves a much better acceleration effect than FlashAttention-2 (FA2) on other general transformers, such as CogvideoX-2B, which uses 30 attention blocks with a constant sequence length $m=17776$ and head embedding dimension $c=64$.
>
> ## Table R3-5. The performance comparison of vanilla ViT architectures and the versions using DuSA2 to replace VSA. All models are trained from scratch on ImageNet-1K. Block size $b$ is 14 for the tiny model and 24 for the large model. Padding is used to make the sequence length divisible by the block size.
> |Backbone|Resolution|Top1 Acc|Param|GFLOPS|FPS (imgs/s)|Inference Memory (GB)|
> |-|-|-|-|-|-|-|
> |ViT-DuSA2-L/16|$384^2$|**77.2**|307M|**174.54**|230|3.0|
> |ViT-L/16|$384^2$|76.5|307M|191.21/174.85 (FA2)|179/**272 (FA2)**|3.7/**2.9 (FA2)**|
> |ViT-DuSA2-T/16|$224^2$|**72.9**|5.8M|**1.07**|10718|**2.0**|
> |ViT-T/16|$224^2$|72.6|5.8M|1.26/1.08 (FA2)|10644/**15350 (FA2)**|2.2/**2.0 (FA2)**|
>
> ---
>
> **References:**
>
> [1] Xu, R., Xiao, G., Huang, H., Guo, J., & Han, S. (2025). Xattention: Block sparse attention with antidiagonal scoring. *arXiv preprint arXiv:2503.16428*.

---

> ### Author Response · Authors · 2025-08-01
>
> There is a small typo in Table R3-5: The term "GFLOPS" should be "GFLOPs". FPS and Inference Memory in Table R3-5 are obtained on a single H20 (96GB) GPU using a batch size of 32 for large (L) models and 512 for tiny (T) models. We are sorry for these unclear points.

---

> ### Author Response · Authors · 2025-08-03
>
> Another two points to clarify is: The number of attention layers in ViT-T/16 is 12 and the number of attention layers in ViT-L/16 is 24. We just wrote it to a 2-2-6-2 (layers number for each level) architecture form (for ViT-T/16) / a 2-2-18-2 (layers number for each level) architecture form (for ViT-L/16) for comparison with the four level architecture of Swin-T/Swin-B.

---

> ### Author Response · Authors · 2025-08-04
>
> The comparison of DuSA2, FlashAttention-2 (FA2), and SpargeAttn [2] on the acceleration of CogvideoX-2B is available in **Section 4.6 and Table 5 of the original manuscript**.
>
> ---
>
> **References:**
>
> [2] Zhang, J., Xiang, C., Huang, H., Wei, J., Xi, H., Zhu, J., & Chen, J. (2025). Spargeattn: Accurate sparse attention accelerating any model inference. *arXiv preprint arXiv:2502.18137*.

---

> ### Author Response · Authors · 2025-08-05
> **Looking Forward to Your Response**
>
> We would greatly appreciate it if you could tell us whether our rebuttal addresses your concerns. If there are no remaining concerns, we would greatly appreciate it if you could consider updating your evaluation of our paper. We welcome any further questions you may have about our work.
>
> We are looking forward to your response.
>
> Thank you for your time and effort in reviewing our paper!
>
> ---
>
> Best regards,
>
> Authors

---

> ### Comment · Reviewer_iq85 · 2025-08-06
>
> Thanks the authors for the response. It solves some of my major concerns. Would be good to see more analysis in future versions.

---

> ### Author Response · Authors · 2025-08-07
> **Thank You for Your Response**
>
> Thank you for your response! We will **add more analysis** including but not limited to **highlight and enrich the analysis of experimentally shown good results, add the analysis on training memory consumption patterns in the above rebuttal to the manuscript, and restructure the manuscript according to your and other reviewers’ constructive comments to make the key design decisions and results of the paper more informative** in the future version. If you have any remaining concerns or other questions about our work please feel free to tell us.
>
> Thank you again for your time and effort in reviewing our paper!
>
> ---
>
> Best regards,
>
> Authors

---

### Official Review · Reviewer_mVoD · 2025-07-02

**Clarity:** 2
**Significance:** 3
**Originality:** 3
**Rating:** 4
**Confidence:** 3

**Summary:**

The authors introduce a new multi-stage sparse attention method to reduce the computational complexity compared to naïve full self attention, while keeping the approximation error to a minimum. The efficacy is validated across different vision and long-sequence experiments.

**Questions:**

**[Q1]:** The models used for DuSA are CSWin and Swin – both models that use convolutional down-sampling in their multi-stage architecture;
$\textrightarrow$  I would be interested if the authors have tried to use their approach in a conventional ‘monolithic’ ViT? Since this keeps the resolution of the feature maps consistent throughout the architecture, I’d assume this provide even more opportunity to improve efficiency given that quadratic complexity of VSA?
$\textrightarrow$  It could also provide ‘cleaner’ insights given the simple structure of the architecture.


**[Q2]:** The models across all experiments seem to be around 50-60M parameters; Have the authors tried different scales of the architectures to validate that the attention works similarly across these, given that larger models can sometimes perform different internal processing steps than small models? How does it work for very small model?

**[Q3]:** While I do appreciate thorough derivations of the proposed methods, I feel like the derivation of the upper bound somewhat stands ‘by itself’ in this paper. After the derivation, the theoretical results are not really used to further analyse or interpret the empirical findings; While there are some empirical findings placed in the appendix in terms of error, they don’t really back up the theoretical ones; Rather, the authors themselves comment that the bounds are probably not tight enough.
$\textrightarrow$  Have the authors thought about better combining these theoretical results with some empirical validation regarding whether the lower approximation errors actually are reflected in the results on actual tasks?
$\textrightarrow$  If not, I’d suggest shortening the bound derivation and using the space instead to provide further insights to the reader in terms of the method, e.g. by including some of the visualisations/analyses currently placed in the appendix – e.g. Fig A2, or the swapped execution of stages (Tab A4), or similar.


**Further Comments / Corrections**:
- While Strategy 1 and 2 are introduced, it is often unclear which one has been used -- and why there are two in the first place; It might be worth providing further analysis in the main paper, including some visualisations that are currently in the appendix (Fig. A2)!
- The authors use ‘hard-aware’ multiple times in their manuscript – I assume they mean “hardware-aware”?
- In Method 3.1 (VSA) l123: The statement “Since m >> c, …” is incorrect in two ways:
  1) The number of patches being much larger than the dimension is not a required condition for the attention-matrix to grow quadratically with m (it does this always, given the nature of its computation)
  2) The statement itself is not necessarily true: Typical ViT-style VSA on ImageNet results in a sequence length of 196+1 tokens ((224/16) x (224/16)) +cls token; but the number of channels is typically of the same scale or larger (192 up to 1280)

**Ethical Concerns:**

["NO or VERY MINOR ethics concerns only"]

**Final Justification:**

Taking the authors' rebuttal, my fellow reviewers' and my own concerns and questions into account, I think this paper is still a borderline case that leans slightly more towards 'accept'; although major restructuring of the paper should be performed by the authors (as recommended) in case this work gets accepted.

**Limitations:**

Some limitations are discussed in the appendix of the paper, but these relate mainly to the tightness of the derived bound. I’d like to recommend the authors to think about some additional limitations inherent to their current approach in terms of applicability and/or other aspects that might be of interest to the reader/practitioner.

**Paper Formatting Concerns:**

No formatting concerns; However, for many questions in the checklist, the authors have simply used the justification “N/A” – i.e. provided no justification at all, although they have answered the question with ‘yes’;
This should be corrected, and proper justifications as well as references to the sections where these questions are answered should be provided!

**Quality:**

3

**Strengths And Weaknesses:**

## Strengths
**Originality & Significance:**
- Novel two-stage method that improves upon other sparse single-stage ones while being still competitive in terms of efficiency/throughput
- Authors demonstrate the applicability via improvements across five different tasks

**Quality:**
- Contributions are generally well-placed within the wider area of research
- Theoretical derivations provided for the underlying concepts

**Clarity:**
- Contributions and underlying motivation clearly outlined in the intro
- Explanation of key-concepts well-supported through a good mix of figures
-	The paper is mostly well written and easy to read and follow

## Weaknesses
- Clarity and insights provided to the reader could be improved -- see questions
- Since the proposed DuSA aims to approximate/replace vanilla scaled dot-product attention, at least one ‘simpler’ architecture that exclusively uses this type of attention on a global scale (e.g. ViT) would provide valuable insights -- see questions
- Somewhat limited experiments in terms of variation in model scale – see questions

---

> ### Author Rebuttal · Authors · 2025-07-30
>
> **We would like to thank the reviewer for the constructive comments provided to help us improve the quality of the paper. We have addressed the issues raised in the comments point-by-point as follows:**
>
> ---
>
> **Response to Q1:** We have applied DuSA2 (DuSA using Strategy 2) to the simple vanilla ViT architectures (ViT-T/16 and ViT-L/16). **Table R2-1** shows the performance comparison of vanilla ViT architectures and the versions using DuSA2 to replace VSA. The sequence length $m$ of vanilla ViT architecture is relatively short; for the resolution of $224^2$, its maximum sequence length is only $196+1$, and its four levels have sequence lengths: $197$-$197$-$197$-$197$ with a constant embedding dimension $c=64$ for each head. Even for the resolution of $384^2$, its maximum sequence length is only $576+1$, and its four levels have sequence lengths: $577$-$577$-$577$-$577$ with a constant embedding dimension $c=64$. The maximum sequence length of Swin and CSWin is $m=3136$/$9216$ for the resolution of $224^2$/$384^2$ and is significantly larger than $197$/$577$. Their four levels have sequence lengths: $3136$-$784$-$196$-$49$/$9216$-$2304$-$576$-$144$ for the resolution of $224^2$/$384^2$ with a constant embedding dimension $c=32$ for each head. As the ablation study of **Table A2 of the original manuscript** shows, if the sequence length $m$ is not significantly larger than the head embedding dimension $c$, the acceleration effect will be limited, which cannot completely show the superiority of our method for acceleration. Although the acceleration effect of DuSA on the ViT series is limited due to the short sequence length, DuSA achieves a much better acceleration effect than FlashAttention-2 (FA2) on other monolithic transformers, such as CogvideoX-2B, which uses 30 attention blocks with a constant sequence length $m=17776$ and head embedding dimension $c=64$.
>
> ## Table R2-1. The performance comparison of vanilla ViT architectures and the versions using DuSA2 to replace VSA. All models are trained from scratch on ImageNet-1K. Block size $b$ is 14 for the tiny model and 24 for the large model. Padding is used to make the sequence length divisible by the block size.
> |Backbone|Resolution|Top1 Acc|Param|GFLOPS|FPS (imgs/s)|Inference Memory (GB)|
> |-|-|-|-|-|-|-|
> |ViT-DuSA2-L/16|$384^2$|**77.2**|307M|**174.54**|230|3.0|
> |ViT-L/16|$384^2$|76.5|307M|191.21/174.85 (FA2)|179/**272 (FA2)**|3.7/**2.9 (FA2)**|
> |ViT-DuSA2-T/16|$224^2$|**72.9**|5.8M|**1.07**|10718|**2.0**|
> |ViT-T/16|$224^2$|72.6|5.8M|1.26/1.08 (FA2)|10644/**15350 (FA2)**|2.2/**2.0 (FA2)**|
>
> ---
>
> **Response to Q2:** We have applied DuSA2 for small models like ViT-T/16, which only has 5.8M parameters, and large models like ViT-L/16, which has 307M parameters. Details are available in the **Response to Q1**. For larger models, please see the acceleration of using DuSA2 on CogvideoX-2B, as **Table 5 of the original manuscript** shows, and using DuSA2 on the acceleration of the prefilling stage of the Llama-3.1-8B-Instruct model, as **Tables R2-2** and **R2-3** show.
>
> ## Table R2-2. The performance comparison of different attention methods on LongBench using the Llama-3.1-8B-Instruct model. Note: Higher scores mean better performance.
> |Method|Full (FlashInfer’s implementation of FlashAttention)|XAttention (stride=8) antidiagonal patterns|DuSA2 (stride=8) antidiagonal patterns|DuSA2 (stride=8) diagonal patterns|DuSA2 (stride=16) antidiagonal patterns|
> |-|-|-|-|-|-|
> |NarrativeQA|31.44|30.48|30.81|31.34|30.21|
> |Qasper|25.07|26.04|24.28|25.96|**27.57**|
> |MultiFieldQA-en|**29.40**|29.28|28.88|28.78|29.00|
> |MultiFieldQA-zh|**61.68**|61.67|58.28|61.65|60.99|
> |HotpotQA|16.89|17.33|**18.09**|16.92|17.34|
> |2WikiMultihopQA|17.00|16.34|16.03|16.35|**17.02**|
> |MuSiQue|11.79|11.88|10.21|11.86|**12.04**|
> |DuReader|34.93|34.64|35.40|35.60|**36.13**|
> |GovReport|34.22|34.60|**34.68**|34.54|34.59|
> |QMSum|23.25|23.24|23.21|**23.43**|23.18|
> |MultiNews|26.69|27.08|26.92|**27.17**|26.96|
> |VCSUM|15.91|16.11|15.77|**16.21**|15.73|
> |TREC|72.50|71.50|72.50|**73.00**|71.50|
> |TriviaQA|91.65|90.97|**92.15**|90.62|90.90|
> |SAMSum|43.74|**44.13**|43.27|43.59|43.67|
> |LSHT|46.00|46.50|**47.00**|46.50|46.00|
> |PassageCount|5.95|5.23|**7.25**|6.38|6.39|
> |PassageRetrieval-en|**98.20**|88.68|96.84|89.26|97.76|
> |PassageRetrieval-zh|**77.11**|74.40|76.67|75.06|73.76|
> |LCC|52.19|53.23|52.28|53.24|**53.36**|
> |RepoBench-P|49.14|**50.94**|48.97|50.16|49.56|
> |Average|**41.18**|40.68|40.92|40.84|41.13|
>
> ## Table R2-3. Speedup relative to FlashAttention (FlashInfer’s implementation) obtained by different attention mechanisms across different sequence lengths using Llama-3.1-8B-Instruct on a single H20 (96GB) GPU.
> |Sequence Length (K)|DuSA2 (stride=8) antidiagonal patterns Speedup|DuSA2 (stride=16) antidiagonal patterns Speedup|XAttention (stride=8) Speedup|
> |-|-|-|-|
> |4|1.02|1.12|**1.93**|
> |8|1.59|2.36|**2.83**|
> |16|2.25|**4.67**|4.05|
> |32|2.79|**6.63**|5.06|
> |64|3.36|**9.11**|6.00|
> |128|3.09|**11.20** |5.57|
>
> ---
>
> **Response to Q3:** We will revise the final version of the paper according to reviewer’s suggestion to shorten the bound derivation and use the space instead to provide further insights to the reader in terms of the method, e.g. by including some of the visualizations/analyses currently placed in the appendix – e.g. **Fig. A2**, or the swapped execution of stages (**Table A4**), or similar.
>
> ---
>
> **Response to Further Comment 1:** We will clarify the use of Strategies 1 and 2, move more visualizations from the appendix to the main body, and shorten the bound derivation.
>
> ---
>
> **Response to Further Comment 2:** We will revise all “hard-aware” to “hardware-aware”.
>
> ---
>
> **Response to Further Comment 3:** Indeed, as the reviewer pointed out, the nature of the scaled dot product attention calculation leads to a complexity of $O(m^2\times c)$. We will make the statement more rigorous. For vanilla ViT, its sequence length is $m=(196+1)$/$(576+1)$ for a resolution of $224^2$/$384^2$, and its embedding dimension is $c=64$ for each head for both $224^2$ and $384^2$ resolutions. Hence, it still satisfies $m > c$. And in the current stage, a long sequence length $m$ is preferred. Take Swin and CSWin for example, their embedding dimension is $c=32$ for each head for all four levels in both $224^2$ and $384^2$ resolutions. Their four levels have sequence lengths: $3136$-$784$-$196$-$49$ (for the input resolution of $224^2$)/$9216$-$2304$-$576$-$144$ (for the input resolution of $384^2$). CogvideoX-2B uses a sequence length $m=17776$ with $30$ heads. Each head has an embedding dimension $c=64$. Llama-3.1-8B-Instruct has $32$ heads. Each head has an embedding dimension $c=128$. Llama-3.1-8B-Instruct supports a maximum sequence length of 128K. $m$ is also significantly larger than $c$. Our method is designed for the acceleration for the case $m \gg c$. The larger $m$ is than $c$, the more obvious the acceleration effect.
>
> **Response to Limitations:** We will add some additional limitations (for example, the acceleration effect of DuSA may be limited when $m$ is not significantly larger than $c$) inherent to our DuSA in terms of applicability or other aspects that might be of interest to the reader or practitioner.
>
> **Response to Paper Formatting Concerns:** We will revise the checklist and provide proper justifications as well as references to the sections where these questions are answered.

---

> > ### Comment · Reviewer_mVoD · 2025-08-04
> > **Thank you for the response**
> >
> > I'd like to thank the authors for the response to my questions, which have been mostly appropriately answered.
> >
> > I do think the manuscript will benefit from some major restructuring (as mentioned in my own and fellow reviewers comments), especially around swapping the theoretical derivation with content that is more informative for the key design decisions and result of the paper.
> >
> > One more question re: the two strategies:
> > I'm still missing the reasoning behind the two strategies S1 and S2 -- is this simply motivated empirically, i.e. the second one is like the first one but more 'extensive' (and therefore costly)? Or is there any different reasoning behind this?

---

> > > ### Author Response · Authors · 2025-08-05
> > > **Reply to the Further Question of Reviewer mVoD by Authors**
> > >
> > > Thank you for your response! We will restructure the manuscript according to your and other reviewers’ constructive comments to make the key design decisions and results of the paper more informative in the next version.
> > >
> > > ---
> > >
> > > **Response to the Further Question:** DuSA1 uses diagonal patterns of the attention matrix to perform block-level mixture, as **Fig. 2 of the original manuscript** shows. Only the tokens corresponding to the positions of the same color (excluding the elements of gray color (zero values)) in the attention matrix $\mathbf{A}'$ are aggregated. DuSA2 further aggregates the attention values of the same position of the diagonal pattern of each color, as **Fig. 3 of the original manuscript** shows. Take **Fig. 3 of the original manuscript** for example, the first attention value ($\mathbf{A}_1'' (1,1)$) in the upper left corner of the orange diagonal pattern, the first attention value ($\mathbf{A}_2'' (1,1) = \mathbf{A}_1''(2,2)$) in the upper left corner of the blue diagonal pattern, the first attention value ($\mathbf{A}_3'' (1,1) = \mathbf{A}_1'' (3,3)$) in the upper left corner of the green diagonal pattern, and the first attention value ($\mathbf{A}_4'' (1,1) = \mathbf{A}_1'' (4,4)$) in the upper left corner of the pink diagonal pattern are aggregated which includes more cross block information and may provide better performance than DuSA1. At the same time, DuSA2 provides a lower approximation error and shows higher performance than DuSA1 after training, as shown in **Fig. A1 and Table A3 of the original manuscript**, which further verifies our assumption. Indeed, DuSA2 has a slightly higher complexity than DuSA1 due to the aggregation of attention values. The rolling indices used to obtain $\mathbf{A}_i''$ ($i\neq1$) can be performed by a simple reshaping operation.
> > >
> > > We welcome any other questions you may have about our work.
> > >
> > > ---
> > >
> > > Best regards,
> > >
> > > Authors

---

> > > > ### Author Response · Authors · 2025-08-05
> > > > **Looking Forward to Your Response**
> > > >
> > > > We would greatly appreciate it if you could tell us whether the above answer addresses your further question. If there are no remaining concerns, we would greatly appreciate it if you could consider updating your evaluation of our paper. We welcome any other questions you may have about our work.
> > > >
> > > > We are looking forward to your response.
> > > >
> > > > Thank you for your time and effort in reviewing our paper!
> > > >
> > > > ---
> > > >
> > > > Best regards,
> > > >
> > > > Authors

---

> > > > > ### Comment · Reviewer_mVoD · 2025-08-08
> > > > > **Thanks for the clarification**
> > > > >
> > > > > I'd like to thank the authors for the additional clarification, which confirms my understanding.
> > > > > Taking my own and my fellow reviewers' reflections into consideration, I'll be maintaining my score for now.

---

> > > > > > ### Author Response · Authors · 2025-08-08
> > > > > > **Thank You for Your Response**
> > > > > >
> > > > > > Thank you for your response!
> > > > > >
> > > > > > Thank you again for your time and effort in reviewing our paper!
> > > > > >
> > > > > > ---
> > > > > >
> > > > > > Best regards,
> > > > > >
> > > > > > Authors

---

> ### Author Response · Authors · 2025-08-01
>
> There is a small typo in Table R2-1: The term "GFLOPS" should be "GFLOPs". FPS and Inference Memory in Table R2-1 are obtained on a single H20 (96GB) GPU using a batch size of 32 for large (L) models and 512 for tiny (T) models. We are sorry for these unclear points.

---

> ### Author Response · Authors · 2025-08-03
>
> Another two points to clarify is: The number of attention layers in ViT-T/16 is 12 and the number of attention layers in ViT-L/16 is 24. We just wrote it to a 2-2-6-2 (layers number for each level) architecture form (for ViT-T/16) / a 2-2-18-2 (layers number for each level) architecture form (for ViT-L/16) for comparison with the four level architecture of Swin-T/Swin-B.

---

### Official Review · Reviewer_87kG · 2025-07-03

**Clarity:** 3
**Significance:** 3
**Originality:** 3
**Rating:** 4
**Confidence:** 4

**Summary:**

This paper proposes the Dual-Stage Sparse Attention (DuSA). DuSA performs two stage sparse attention mechanism where in the first it performs intra block to learn local inductive biases and in second stage they learn interblock attention mechanism to learn long range dependencies. this two stage mechanism reduces the error in vanilla scaled-dot product attention compared to basic single-stage sparse mechanisms. The first stage  obtains  the local inductive biases and the second stage aggregates all blocks according to their similarities  to aggregate  long-range dependencies.

**Questions:**

Questions:
- even though DuSA has less memory, DuSA has equal FLOPs or more compared to existing methods like ELFATT.
- Can DuSA be extended auto-regressive based methods?

**Ethical Concerns:**

["NO or VERY MINOR ethics concerns only"]

**Final Justification:**

Thankyou Authors for providing rebuttal. After reading the reviews from other reviewers and authors responses I would like to maintain my initial rating.

**Limitations:**

Limitations are discussed in the paper

**Paper Formatting Concerns:**

no concerns

**Quality:**

3

**Strengths And Weaknesses:**

Strengths:
- DuSA has low error in approximating vanilla scaled-dot product attention
- Both stages have low computational complexity and are compatible with memory efficient methods, and can be further accelerated.  DuSA can further improve and accelerate some advanced sparse attention mechansims
- the paper clearly explained the math behind the approximation process of the two stage process in DuSA.
- DuSA is compared with state-of-the-art attention mechanisms in LRA.  DuSA conducted experiments in image classification, semantic segmentation and object detection performance

Weaknesses:
- ELFATT performance is same or better than DuSA in tables 3 and 4. ELFATT has less number of FLOPs compared DuSA

---

> ### Author Rebuttal · Authors · 2025-07-30
>
> **We would like to thank the reviewer for the constructive comments provided to help us improve the quality of the paper. We have addressed the issues raised in the comments point-by-point as follows:**
>
> ---
>
> **Response to W1:** ELFATT [1] is a SOTA for ViT acceleration, which was just online several months before DuSA. The FLOPs of DuSA are lower than ELFATT. You may have confused ELFATT with EFFATT [2]. EFFATT is a kernelized linear method, and its theoretical complexity is indeed lower than that of DuSA. But its performance is significantly inferior to DuSA. Its real speed doesn’t show a significant superiority compared to DuSA. Because both stages of DuSA can be further sped up by memory-efficient methods.
>
> ---
>
> **Response to Q1:** The number of floating point operations (FLOPs) doesn’t represent the real speed of the method, as pointed out by [1,3,4]. Even though some methods have lower FLOPs, their real speed could be slower. Hence, we provide the inference throughput of all methods, which will be more accurate to represent the real speed of each method.
>
> ---
>
> **Response to Q2:** Yes, it can. XAttention [5] has shown that the diagonal pattern-based attention mechanisms can be used in accelerating the prefilling stage for recent auto-regressive transformer-based (decoder-only) LLMs. Based on XAttention, we introduce DuSA2 (Strategy 2: Rolling indices-based attention scores fusion) to search antidiagonal or diagonal patterns of attention matrices to filter important or unimportant blocks for further dynamic sparse attention only performed within important blocks. As **Table R1-1** shows, compared to XAttention, the performance of DuSA2 is closer to that of full attention. As **Tables R1-1** and **R1-2** show, DuSA2 can use a larger stride (larger sparsity) to achieve similar or higher performance than XAttention, further accelerating full attention. By introducing antidiagonal patterns, the performance of DuSA2 can be further improved.
>
> ## Table R1-1. The performance comparison of different attention mechanisms on LongBench using the Llama-3.1-8B-Instruct model. Note: Higher scores mean better performance.
> |Method|Full (FlashInfer’s implementation of FlashAttention)|XAttention (stride=8)|DuSA2 (stride=8) antidiagonal patterns|DuSA2 (stride=8) diagonal patterns|DuSA2 (stride=16) antidiagonal patterns|
> |-|-|-|-|-|-|
> |NarrativeQA|31.44|30.48|30.81|31.34|30.21|
> |Qasper|25.07|26.04|24.28|25.96|**27.57**|
> |MultiFieldQA-en|**29.40**|29.28|28.88|28.78|29.00|
> |MultiFieldQA-zh|**61.68**|61.67|58.28|61.65|60.99|
> |HotpotQA|16.89|17.33|**18.09**|16.92|17.34|
> |2WikiMultihopQA|17.00|16.34|16.03|16.35|**17.02**|
> |MuSiQue|11.79|11.88|10.21|11.86|**12.04**|
> |DuReader|34.93|34.64|35.40|35.60|**36.13**|
> |GovReport|34.22|34.60|**34.68**|34.54|34.59|
> |QMSum|23.25|23.24|23.21|**23.43**|23.18|
> |MultiNews|26.69|27.08|26.92|**27.17**|26.96|
> |VCSUM|15.91|16.11|15.77|**16.21**|15.73|
> |TREC|72.50|71.50|72.50|**73.00**|71.50|
> |TriviaQA|91.65|90.97|**92.15**|90.62|90.90|
> |SAMSum|43.74|**44.13**|43.27|43.59|43.67|
> |LSHT|46.00|46.50|**47.00**|46.50|46.00|
> |PassageCount|5.95|5.23|**7.25**|6.38|6.39|
> |PassageRetrieval-en|**98.20**|88.68|96.84|89.26|97.76|
> |PassageRetrieval-zh|**77.11**|74.40|76.67|75.06|73.76|
> |LCC|52.19|53.23|52.28|53.24|**53.36**|
> |RepoBench-P|49.14|**50.94**|48.97|50.16|49.56|
> |Average|**41.18**|40.68|40.92|40.84|41.13|
>
> ## Table R1-2. Speedup relative to FlashAttention (FlashInfer’s implementation) obtained by different attention mechanisms across different sequence lengths using Llama-3.1-8B-Instruct on a single H20 (96GB) GPU.
> |Sequence Length (K)|DuSA2 (stride=8) antidiagonal patterns Speedup|DuSA2 (stride=16) antidiagonal patterns Speedup|XAttention (stride=8) Speedup|
> |-|-|-|-|
> |4|1.02|1.12|**1.93**|
> |8|1.59|2.36|**2.83**|
> |16|2.25|**4.67**|4.05|
> |32|2.79|**6.63**|5.06|
> |64|3.36|**9.11**|6.00|
> |128|3.09|**11.20** |5.57|
>
> ---
>
> **References:**
>
> [1] Wu, C., Che, M., Xu, R., Ran, Z., & Yan, H. (2025). ELFATT: Efficient linear fast attention for vision transformers. *arXiv preprint arXiv:2501.06098*.
>
> [2] Shen, Z., Zhang, M., Zhao, H., Yi, S., & Li, H. (2021). Efficient attention: Attention with linear complexities. In *Proceedings of the IEEE/CVF winter conference on applications of computer vision* (pp. 3531-3539).
>
> [3] Chen, J., Kao, S. H., He, H., Zhuo, W., Wen, S., Lee, C. H., & Chan, S. H. G. (2023). Run, don't walk: chasing higher FLOPS for faster neural networks. In *Proceedings of the IEEE/CVF conference on computer vision and pattern recognition* (pp. 12021-12031).
>
> [4] Zhang, J., Xiang, C., Huang, H., Wei, J., Xi, H., Zhu, J., & Chen, J. (2025). Spargeattn: Accurate sparse attention accelerating any model inference. *arXiv preprint arXiv:2502.18137*.
>
> [5] Xu, R., Xiao, G., Huang, H., Guo, J., & Han, S. (2025). Xattention: Block sparse attention with antidiagonal scoring. *arXiv preprint arXiv:2503.16428*.

---

> ### Author Response · Authors · 2025-08-05
> **Thank You for Submitting the Final Rating**
>
> Thank you for submitting the final rating! We would greatly appreciate it if you could tell us whether our rebuttal addresses your concerns. If there are no remaining concerns, we would greatly appreciate it if you could consider updating your evaluation of our paper. We welcome any further questions you may have about our work.
>
> We are looking forward to your response.
>
> Thank you for your time and effort in reviewing our paper!
>
> ---
>
> Best regards,
>
> Authors

---

> > ### Author Response · Authors · 2025-08-06
> > **The Final Justification and Rating Are Not Visible to Authors**
> >
> > Dear **Reviewer 87kG**,
> >
> > The final justification and rating are not visible to authors. We would greatly appreciate it if you could tell us whether our rebuttal addresses your concerns. If there are no remaining concerns, we would greatly appreciate it if you could consider updating your evaluation of our paper. If you have any further questions about our work, please feel free to tell us.
> >
> > Thank you for your time and effort in reviewing our paper!
> >
> > ---
> >
> > Best regards,
> >
> > Authors

---

> > > ### Comment · Reviewer_87kG · 2025-08-06
> > > **Response to the rebuttal**
> > >
> > > Thankyou Authors for providing response to my concerns., for now I am maintaining my score. I will revisit and finalize my score after considering the perspectives of the other reviewers and seeing whether there are additional points that might warrant a further adjustment.

---

> > > > ### Author Response · Authors · 2025-08-06
> > > > **Thank You for Your Response**
> > > >
> > > > Thank you for your response!
> > > >
> > > > Thank you again for your time and effort in reviewing our paper!
> > > >
> > > > ---
> > > >
> > > > Best regards,
> > > >
> > > > Authors

---

### Note · Authors · 2025-08-14

**We express our gratitude to all reviewers for their thoughtful and constructive feedback.**

---

**Highlighted strengths noted by reviewers:**

1. Superior performance and more significant speedup effects over SOTA baselines, clear motivation, and strong novelty (**all four reviewers**).

2. Clear derivation of the two-stage sparse attention approximation (**Reviewers 87kG, mVoD, and iq85**).

3. Strong cross-task generalization (**all four reviewers**).

---

**Main concerns:**

1. The current (loose) theoretical bound weakens the relation to experiments: several reviewers suggest swapping the placement of theory of the main text and some experimental analysis of the appendix (**Reviewers mVoD, iq85, and Ah1n**).

2. Need for more validation on simpler or more general Transformer architectures (**Reviewers mVoD and iq85**).

3. Need for more analysis on memory consumption patterns during training (**Reviewer iq85**).

4. Noticeable accuracy drops in the plug-and-play setting (**Reviewers iq85 and Ah1n**).

---

**Planned revisions:**

1. Reorganize the paper: move detailed theory to the appendix and bring more experimental analysis to the main text (**this is a minor issue and can be fixed quickly and properly**).

2. Add results and analysis on simpler architectures (**already provided in the rebuttal; will be included in the revision**).

3. Add results and analysis on memory consumption patterns during training (**already provided in the rebuttal; will be included in the revision**).

4. Clarify scope of DuSA as a static sparse attention method for accelerating training and inference; after training it matches or exceeds VSA. We also implement a higher-performing plug-and-play variant inspired by XAttention [1] (**already provided in the rebuttal; will be included in the revision**).

---

**We appreciate the reviewers’ recognition of our work’s potential to advance sparse attention research and believe the paper merits publication to stimulate further discussion.**

---

**References:**

[1] Xu, R., Xiao, G., Huang, H., Guo, J., & Han, S. (2025). XAttention: Block sparse attention with antidiagonal scoring. *arXiv preprint arXiv:2503.16428*.

---

### Decision · Program_Chairs · 2025-09-17

**Decision:**

Accept (poster)

**Comment:**

This paper proposes DuSA, a dual-stage sparse attention mechanism that is empirically strong across tasks and shows practical speedups while remaining compatible with memory-efficient implementations. Reviewers highlight clear motivation and derivations and broad cross-task validation, with the method improving upon single-stage sparse baselines in accuracy/throughput trade-offs. The reviewers find the contribution is novel and practically relevant, with strong evidence across diverse benchmarks; remaining issues are editorial/clarificatory rather than technical. Thus, the AC decides to accept the submission.